# Selective and regulated trapping of nicotinic receptor weak base ligands and relevance to smoking cessation

Anitha P Govind[1], Yolanda F Vallejo[2], Jacob R Stolz[3], Jing-Zhi Yan[3], Geoffrey T Swanson[3], William N Green[1,4]*

[1]Department of Neurobiology, University of Chicago, Chicago, United States; [2]National Institute of Dental and Craniofacial Research at the National Institutes of Health, United States; [3]Department of Pharmacology, Northwestern University, Feinberg School of Medicine, Evanston, United States; [4]Marine Biological Laboratory, Woods Hole, United States

**Abstract** To better understand smoking cessation, we examined the actions of varenicline (Chantix) during long-term nicotine exposure. Varenicline reduced nicotine upregulation of $\alpha4\beta2$-type nicotinic receptors ($\alpha4\beta2$Rs) in live cells and neurons, but not for membrane preparations. Effects on upregulation depended on intracellular pH homeostasis and were not observed if acidic pH in intracellular compartments was neutralized. Varenicline was trapped as a weak base in acidic compartments and slowly released, blocking $^{125}$I-epibatidine binding and desensitizing $\alpha4\beta2$Rs. Epibatidine itself was trapped; $^{125}$I-epibatidine slow release from acidic vesicles was directly measured and required the presence of $\alpha4\beta2$Rs. Nicotine exposure increased epibatidine trapping by increasing the numbers of acidic vesicles containing $\alpha4\beta2$Rs. We conclude that varenicline as a smoking cessation agent differs from nicotine through trapping in $\alpha4\beta2$R-containing acidic vesicles that is selective and nicotine-regulated. Our results provide a new paradigm for how smoking cessation occurs and suggest how more effective smoking cessation reagents can be designed.

*For correspondence: wgreen@ uchicago.edu

**Competing interests:** The authors declare that no competing interests exist.

## Introduction

Tobacco continues to be widely used world-wide, primarily via cigarette smoking, and is the leading cause of preventable deaths in the United States (*National Center for Chronic Disease Prevention and Health Promotion (US) Office on Smoking and Health, 2014*). The currently approved treatments for smoking cessation are nicotine replacement therapy, bupropion, and varenicline (Chantix). While varenicline is the most effective, successful quit rates only reach ~50% of smokers (*Agboola et al., 2015*). Other therapies or novel approaches are clearly needed to increase rates of smoking cessation, and the design of smoking cessation reagents would be greatly aided with a mechanistic understanding of how the reagents act.

Nicotine binds to high-affinity nicotinic acetylcholine receptors (nAChRs) in the brain, and binding initiates its addictive effects. nAChRs are members of the Cys-loop family of ligand-gated ion channels, all of which are pentameric neurotransmitter receptors (*Karlin, 2002*; *Albuquerque et al., 2009*). In the mammalian CNS, these critical binding sites are nAChRs composed of $\alpha2$ - $\alpha6$ and $\beta2$ - $\beta4$ subunits; the most predominant contains $\alpha4$ and $\beta2$ subunits (*Lindstrom, 1996*; *McGehee and Role, 1995*). The $\alpha4\beta2$ nAChR subtype ($\alpha4\beta2$R) is closely linked to nicotine addiction (*Govind et al., 2009*; *Vezina et al., 2007*; *Lewis and Picciotto, 2013*). Loss of either subunit in $\alpha4$ or $\beta2$ subunit knockout mice reduces the pharmacological and behavioral effects of nicotine (*Picciotto et al., 1998*; *Marubio et al., 2003*). In addition, targeted expression of $\beta2$ subunits in the brain ventral

**eLife digest** Tobacco continues to be widely used worldwide, primarily via cigarette smoking, and is a leading cause of preventable deaths. Stopping smoking is difficult because the nicotine in tobacco is highly addictive, and so several drugs have been developed to help people break their addiction. Varenicline (also known by the trade name Chantix) is a commonly prescribed anti-smoking drug, but it is not fully understood how it works.

Nicotine affects the brain by binding to proteins called nicotinic acetylcholine receptors (nAChRs) that sit on the surface of neurons. This binding releases a number of chemical signals, including some that produce feelings of pleasure. Over time, the receptors become less sensitive to nicotine and produce more "high-affinity" binding sites for nicotine to bind to. This adaptation is one reason why stopping smoking can produce strong feelings of withdrawal.

Previously, varenicline was thought to partially activate nAChRs, preventing nicotine from binding to the receptors. However, this can only explain how varenicline counteracts the rapid-acting effects of nicotine, not nicotine's longer-term effects. Furthermore, it was not known how nAChR signaling responds to long-term exposure to a combination of both drugs (as occurs when people try to quit smoking with the aid of varenicline).

Now, Govind et al. reveal how varenicline reverses the effect of long-term nicotine exposure on nAChR signaling. Both varenicline and nicotine accumulate in acidic compartments – called vesicles – inside cells, where they become charged and less able to move through the cell membrane. When the vesicles also contain high-affinity nAChRs, varenicline becomes trapped inside them and is only slowly released. By contrast, nicotine is not trapped because it exits the vesicles more rapidly. Long-term exposure to nicotine greatly increased the number of vesicles that contained high-affinity nAChRs, thereby trapping more varenicline.

One consequence of trapping varenicline was that the activity of the nAChRs on the surface of the neuron was diminished, apparently through the slow release of the trapped varenicline from the acidic vesicles. This slow release causes the receptors to enter a "desensitized" state in which they do not signal.

Understanding how varenicline counteracts the long-term effects of nicotine on nAChR signaling will help us to design more effective anti-smoking drugs. Govind et al. also found that compounds similar to varenicline become trapped in vesicles, but it is not clear how the degree of trapping of a compound correlates with how effectively it combats nicotine addiction. The results may also help us to understand and treat addictions to other drugs of abuse, such as opioids, amphetamines and cocaine, which have chemical properties that mean they might also be selectively trapped in acidic vesicles.

tegmental area (VTA) of $\beta$2-knockout mice rescues nicotine-seeking behavior and nicotine-induced dopamine release (*Maskos et al., 2005*).

Varenicline is a high-affinity partial agonist for $\alpha4\beta$2R. The rationale for its design as a smoking cessation reagent was predicated on the idea that a partial agonist could compete for and reduce cell-surface $\alpha4\beta$2R activation by nicotine, reduce dopamine overflow in the mesolimbic reward system and thereby reduce the reward sensation of smoking (*Rollema et al., 2007*). Varenicline does indeed reduce the rapid effects of nicotine on $\alpha4\beta$2Rs in VTA dopaminergic neurons and acutely reduces nicotine-induced dopamine release in the nucleus accumbens (*Rollema et al., 2007*). How varenicline alters the longer-lasting effects of nicotine is less clear, however. Nicotine upregulation of nAChRs is linked to different processes in nicotine addiction, including sensitization (*Govind et al., 2009*; *Vezina et al., 2007*) and withdrawal (*De Biasi et al., 2011*) and is the only effect of nicotine on nAChRs that lasts longer than a few minutes. Upregulation occurs when nicotine exposure increases high-affinity nicotine-binding sites in brain, measured by radiolabeled agonists such as nicotine (*Marks et al., 1983*; *Schwartz and Kellar, 1983*; *Benwell et al., 1988*; *Breese et al., 1997*) or epibatidine (*Perry et al., 1999*). Chronic varenicline exposure in mice induces upregulation to the same degree, or even more than, chronic nicotine exposure when measured using $^3$H-epibatidine binding to membrane preparations from different brain regions

(*Hussmann et al., 2012*; *Turner et al., 2011*). Recent studies using PET imaging in humans found that reversal of α4β2R upregulation was correlated with less smoking relapse (*Brody et al., 2013*; *Staley et al., 2006*; *Cosgrove et al., 2009*). Thus, the observation that varenicline induced upregulation like nicotine was surprising and difficult to reconcile with its smoking cessation actions.

In this study, we describe novel, long-lasting changes in nicotine upregulation by varenicline and other α4β2R ligands. The weak base nature of varenicline and other α4β2R ligands was fundamental to this action on upregulation. Nicotinic receptor ligands that are weak bases have two states, the membrane permeable uncharged state and the membrane impermeable protonated state. Nicotine accumulates in intracellular acidic compartments of cells and neurons because it is more highly protonated at the lower pH (*Brown and Garthwaite, 1979*; *Bhagat, 1970*; *Putney and Borzelleca, 1971*), which leads to 'trapping' in these compartments in *Xenopus* oocytes expressing α4β2Rs. After removal of extracellular nicotine, nicotine slowly leaks from the intracellular compartments and causes α4β2R desensitization at the plasma membrane (*Jia et al., 2003*). However, an analogous phenomenon was not observed in mammalian cells heterologously expressing α4β2Rs (*Jia et al., 2003*), raising questions as to its relevance to neuronal adaptations to chronic nicotine exposure. Here we find that varenicline, as well as lobeline and epibatidine, are trapped as weak bases within acidic vesicles of live mammalian cells and neurons; in contrast, nicotine rapidly exits these vesicles. The trapping is selective depending on ligand $pK_a$ and affinity for α4β2Rs in acidic vesicles. Using [125]I-epibatidine binding, we directly measured trapping and slow release from acidic vesicles and found it required α4β2R expression and increased with nicotine upregulation. Using pH-sensitive, pHluorin-tagged α4 subunits, we found that nicotine increased the numbers of acidic vesicles containing α4β2Rs, thereby promoting accumulation of [125]I-epibatidine. Our results provide a new paradigm for how smoking cessation occurs and suggests how more effective smoking cessation reagents can be designed.

## Results

### Varenicline and lobeline reduce nicotine upregulation assayed by [125]I-epibatidine binding sites

We determined if the smoking cessation drug and partial agonist varenicline, as well as other nAChR ligands, caused upregulation of nAChRs or altered nicotine upregulation using HEK cells stably expressing α4β2Rs or cortical neurons expressing endogenous α4β2Rs. α4β2R upregulation was assayed with two approaches: [125]I-epibatidine binding (*Figure 1*) or patch-clamp recording of ACh-evoked currents (*Figure 2*). Binding and function were assayed following 17–20 hr treatments of cells with control media, nAChR ligand (e.g., varenicline), nicotine (1 or 10 μM), and nAChR ligand co-incubated with nicotine. All results were normalized to the amount of binding or mean current amplitude observed with the nicotine-only condition (i.e., the upregulated state). Values for mean, variance and numbers of measures are given in the figure legends.

Varenicline (*Figure 1A*) at 30 μM did not upregulate [125]I-epibatidine binding in α4β2R-expressing HEK cells (*Figure 1B*, 'Var', red columns in the left, 'live cells' half of the graph) nor in cortical neurons (*Figure 1F*, left) following 17 hr of exposure. However, a full concentration-response curve with chronic varenicline revealed an inverted U-shaped dependence of upregulation of binding that peaked at 60–70% of that observed with nicotine, which occurred at varenicline concentrations between 100 nM and 1 μM (*Figure 1—figure supplement 1A*, black circles). When co-applied with nicotine, varenicline (30 μM) prevented nicotine upregulation of binding in both heterologous cells and cortical neurons (*Figure 1B,F*). The reversal of the actions of nicotine on α4β2R binding also was concentration-dependent, with upregulation decreasing exponentially with increased varenicline concentration (*Figure 1—figure supplement 1B*, black squares). Thus, chronic varenicline exposure precludes nicotine upregulation of binding when assayed in living cells.

Recent studies indicate that varenicline concentrations in the brain are likely to be higher than in plasma because of the presence of high-affinity binding to nAChRs (*Rollema et al., 2010*). Based on this study's estimates, unbound varenicline concentrations in the brain fall in the range of 32 to 131 nM and plasma concentrations are approximately 4-fold lower. In addition, a previous study found differences in varencilcine binding to human α4β2Rs compared to rat α4β2Rs when expressed in Xenopus oocytes (*Papke et al., 2010*). In order to compare, human and rat α4β2Rs at relevant

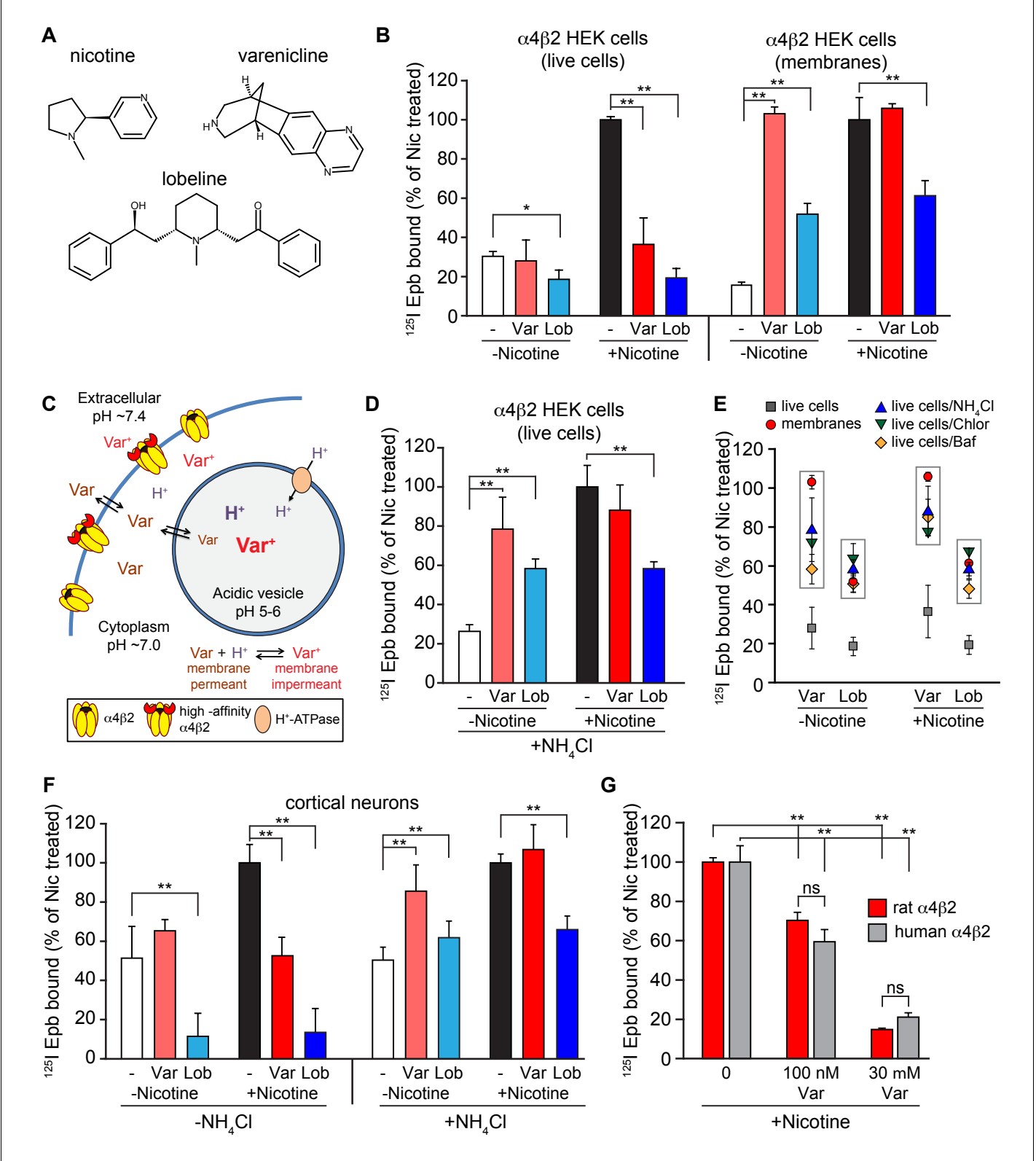

**Figure 1.** Effect of smoking cessation reagents on α4β2R upregulation. (A) Nicotine, varenicline and lobeline chemical structures. (B) Varenicline and lobeline reduced nicotine upregulation of α4*β*2Rs for live cells, but not membranes. $^{125}$I-epibatidine binding performed on live α4*β*2R-expressing HEK cells (left) (n = 3) or membranes (right) (n = 3). Cells were treated for 17 hr with 30 μM varenicline or 30 μM lobeline with or without 10 μM nicotine. Specific epibatidine binding was represented as % of binding relative to nicotine upregulated cells (C) Model of varenicline (Var) trapping in acidic

*Figure 1 continued on next page*

*Figure 1 continued*

vesicles. Varenicline is trapped when protonated in the acidic vesicle lumen.$\alpha 4\beta 2$Rs on the plasma membrane are depicted in two states: $\alpha 4\beta 2$Rs with and without high-affinity binding sites consistent with the findings of Vallejo *et al.* (*Benowitz et al., 2009*). (D) An intracellular acidic compartment is required for varenicline and lobeline effects on upregulation. [125]I-epibatidine binding performed on live $\alpha 4\beta 2$R HEK cells with 20 mM $NH_4Cl$ treatment for 10 min. Cells were exposed to 30 μM varenicline or 30 μM lobeline with or without 10 μM nicotine for 17 hr as in (B) (n = 4). (E) A distribution plot comparing the reduction in upregulation ([125]I-epibatidine binding) by varenicline and lobeline to the recovery after disruption of intracellular pH gradient by membrane preparation or various agents that raise pH in intact cells. Each point represents the mean and the standard error of the mean (s.e.m) from the indicated columns in *Figure 1B,D*, and *Figure 1—figure supplement 2A,B*. (F) [125]I-epibatidine binding on live cortical neurons without (left) or with (right) 20 mM $NH_4Cl$ treatment. Neurons were exposed to varenicline or lobeline as in B) in the presence or absence of 1 μM nicotine (n = 3). (G) Varenicline reduced nicotine induced upregulation of human $\alpha 4\beta 2$Rs. [125]I-epibatidine binding was performed on live cells either stably expressing rat $\alpha 4\beta 2$R or transiently expressing human $\alpha 4\beta 2$Rs. HEK cells were transfected with human $\alpha 4$ and $\beta 2$ subunits for 24 hr. Cells were treated with 100 nM or 30 μM varenicline in the presence of 10 μM nicotine for 17 hr prior to [125]I-epibatidine binding (n = 3). In all the column graphs in (B, D, F, G): *p<0.05; **p<0.00 one by one-way ANOVA with Bonferroni's multiple comparison test; n indicates number of independent experiments performed on separate days and cultures. Columns represent group mean and error bars are the standard error of the mean (s.e.m).

The following figure supplements are available for figure 1:

**Figure supplement 1.** Dose dependence of varenicline and lobeline effects on upregulation.

**Figure supplement 2.** Altering intracellular pH of acidic vesicles alters the effects of varenicline and lobeline.

**Figure supplement 3.** Release by $NH_4Cl$ treatment of $\alpha 4\beta 2$R ligands trapped inside the acidic compartment.

concentrations, we measured the effect of varenicline on nicotine upregulation at 0, 100 nM and 30 μM. We found that at 100 nM varenicline nicotine upregulation was significantly reduced to ~60% of the level without varencline (*Figure 1G*) for both rat and human $\alpha 4\beta 2$Rs. These results indicate that varenicline should reduce nicotine upregulation at concentrations expected to exist in brain. In contrast, varenicline is unlikely to act as a partial agonist at these concentrations (*Rollema et al., 2010*). We also found no significant differences between rat and human $\alpha 4\beta 2$Rs at 0, 100 nM and 30 μM varenicline (*Figure 1G*).

Another nAChR partial agonist, lobeline, reported to have smoking cessation activity (*Damaj et al., 1997*; *Miller et al., 2003*) had effects similar to those of varenicline (*Figure 1A*). Chronic exposure to lobeline alone or together with nicotine reduced [125]I-epibatidine binding levels below that observed for the untreated cells and neurons ('Lob', blue columns in the left, 'live cells' half of the graphs in *Figure 1B,F*). The inverted U-shaped concentration-dependence of lobeline effects on upregulation was similar to that of varenicline. [125]I-epibatidine binding peaked at ~50% of nicotine-induced upregulation, was maximal at 100 nM – 1 μM, (*Figure 1—figure supplement 1C*, black circles), and co-incubation with nicotine reduced upregulation exponentially with increasing concentrations of lobeline (*Figure 1—figure supplement 1D*, black squares). Lobeline therefore has very similar effects as varenicline in preventing nicotine upregulation.

Intact HEK cells or neurons were required to observe the block by varenicline and lobeline of nicotine upregulation. If [125]I-epibatidine binding was performed on membrane fragments instead of intact cells, varenicline upregulated [125]I-epibatidine binding in $\alpha 4\beta 2$R-expressing HEK cells to the same degree as nicotine and did not reduce nicotine upregulation (*Figure 1B*, red columns on right, 'membranes' half of the graph). Lobeline exposure upregulated [125]I-epibatidine binding to about 50% of that of observed with nicotine treatment and reduced nicotine upregulation by 50% (*Figure 1B*, blue columns on right half of the graph). The effects of varenicline and lobeline on nicotine upregulation were therefore distinct in live cells and membrane fragments, which could explain why previous studies found that chronic varenicline exposure upregulated [3]H-epibatidine binding to brain tissue to the same degree as nicotine and reported no effect of varenicline exposure on nicotine upregulation (*Hussmann et al., 2012*; *Turner et al., 2011*). These experiments were done with membrane preparations, whereas the suppressive activity of varenicline on nicotine upregulation was only observed in intact cells.

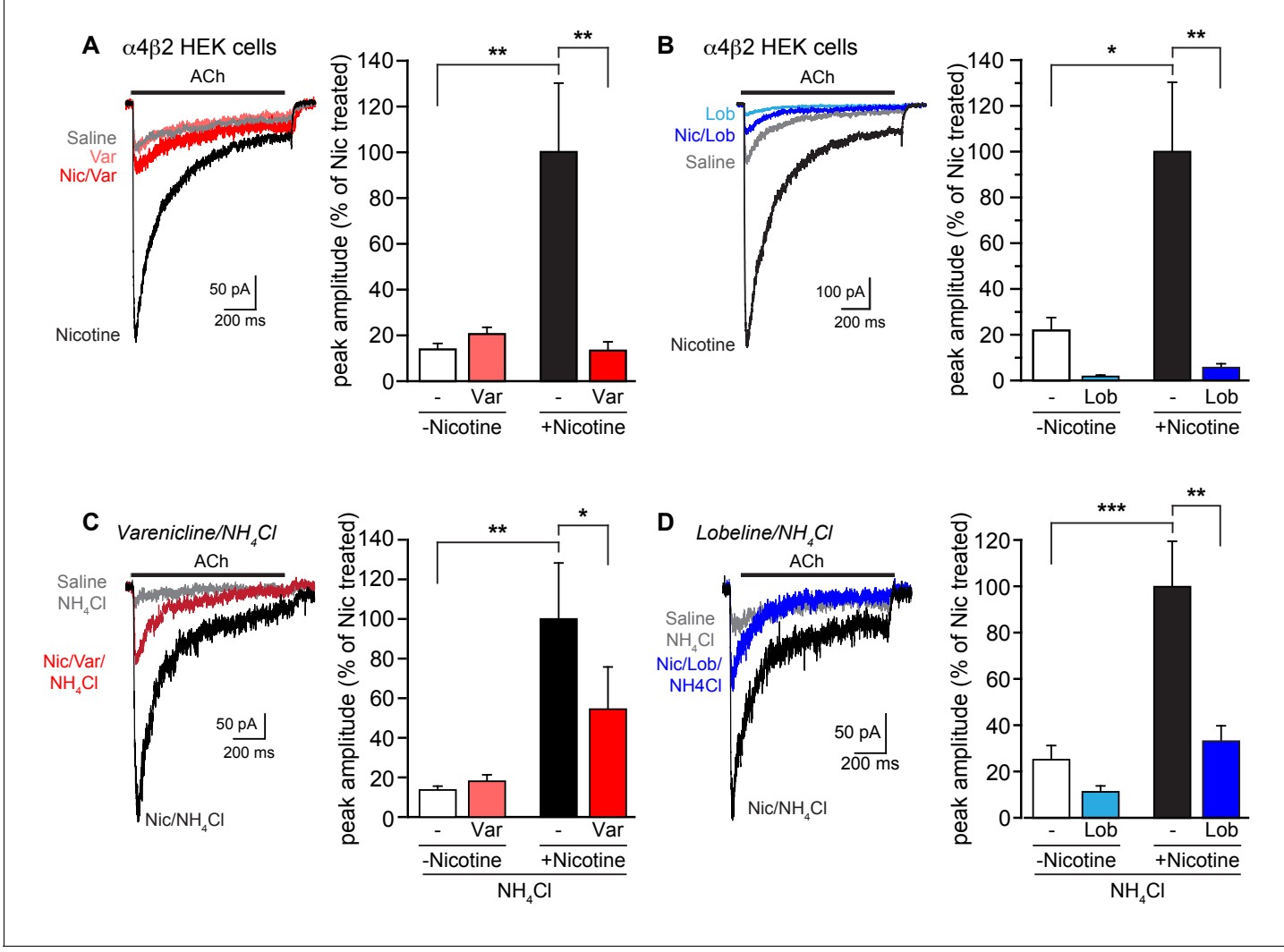

**Figure 2.** Effect of smoking cessation reagents on functional upregulation. (**A**) In this and other recordings, a 17–20 hr treatment with nicotine (Nic, 10 μM) induced a robust, ~5 fold, increase in peak current amplitudes evoked by 1 mM ACh from α4β2R-expressing HEK cells. Varenicline (Var, 30 μM) treatment for an equivalent time had no effect on ACh-evoked peak current amplitudes alone but prevented upregulation of nAChR function when co-incubated with nicotine. Traces represent currents evoked by ACh in cells that were untreated or treated for 17 hr with vehicle, nicotine, varenicline, or with varenicline and nicotine. The number of recordings were 16, 18, 18, and 17, respectively. (**B**) Chronic lobeline (Lob, 30 μM) exposure reduced the peak ACh current amplitude relative to control and prevent nicotine upregulation. Traces represent currents evoked by ACh in cells that were untreated or treated for 17 hr with vehicle, nicotine, lobeline, or with lobeline and nicotine. The number of recordings were 13, 18, 13, and 15, respectively. (**C**) NH₄Cl treatment (two 10 min washes) partially reverses the suppressive effects of varenicline on nicotine-induced functional upregulation. Traces show representative currents evoked by ACh from cells pretreated for 17–20 hr with vehicle, nicotine, or nicotine and varenicline before exposure to NH₄Cl. The varenicline-alone condition is omitted for clarity. The graph is as in (**A**) except that the cells were treated with NH₄Cl as noted. The profound reduction in nicotine-induced upregulation caused by varenicline co-incubation was lessened following treatment with NH₄Cl. The number of recordings were 21, 22, 25, and 25, respectively. (**D**) Traces show representative currents evoked by ACh from cells pretreated for 17–20 hr with vehicle, nicotine, or nicotine and lobeline before exposure to NH₄Cl. In this dataset, lobeline-induced suppression of nicotine upregulation was modestly attenuated by treatment with NH₄Cl. The number of recordings were 14, 15, 16, and 19, respectively. In all the column graphs: * p<0.05; **p<0.01; ***p<0.001 by one-way ANOVA with Tukey's multiple comparison test. Columns show group mean and error bars are the s.e.m.

## An intracellular acidic compartment is required for varenicline and lobeline effects on upregulation and ACh-activated currents

An attractive hypothesis to explain why intact cells were required for suppression of upregulation is that varenicline and lobeline, as weak bases, are concentrated and trapped in intracellular acidic compartments where they are highly protonated. A schematic of how this might occur is displayed

in *Figure 1C*. This process was described previously for nicotine in *Xenopus* oocytes expressing α4β2Rs (*Jia et al., 2003*); furthermore, slow release of nicotine from acidic compartments caused α4β2R desensitization, thereby reducing ACh-induced α4β2R currents. We postulated that similar processes could account for the differential effects of varenicline and lobeline exposure on nicotine-induced upregulation in live cells versus membrane fragments (*Figure 1B*). That is, varenicline and lobeline also could be trapped due to protonation, slowly leak out after removal from the extracellular solution, and consequently reduce $^{125}$I-epibatidine binding and functional upregulation as shown in *Figures 1* and *2*.

To test this hypothesis, we determined if suppression of nicotine upregulation by varenicline and lobeline was relieved by increasing the pH of intracellular compartments, thereby reducing the protonation of the two nAChR ligands. The pH gradient was made more basic using three different strategies. First, live HEK cells were incubated briefly with ammonium chloride ($NH_4Cl$; 20 mM) prior to $^{125}$I-epibatidine binding. We reasoned that this should result in $^{125}$I-epibatidine binding similar to that observed with the membrane preparation in *Figure 1B* (right half) as a result of shifting the equilibrium to uncharged species of the ligands at more basic pHs. $NH_4Cl$ treatment altered the effects of varenicline and lobeline on $^{125}$I-epibatidine binding; both ligands produced upregulation of binding (*Figure 1D* left half, *Figure 1—figure supplement 1A,C*), in contrast to their actions at physiological pH (*Figure 1B*, leftmost trio of columns). Moreover, $NH_4Cl$ exposure following co-incubation with nicotine relieved the suppression of upregulation by varenicline and lobeline (*Figure 1D*, right trio of columns, *Figure 1*-figure supplement B, D), such that $^{125}$I-epibatidine binding in these conditions were indeed similar to the results using membrane preparations (*Figure 1B*, rightmost trio of columns). $NH_4Cl$ treatment of cultured cortical neurons had similar effects on $^{125}$I-epibatidine binding after varenicline or lobeline exposures (*Figure 1F*, right half of graph). We also obtained similar $^{125}$I-epibatidine binding results with HEK cells using two other methods to reduce the pH gradient - incubation with the proton pump inhibitor bafilomycin A (Baf, 50 nM; *Figure 1E*, *Figure 1—figure supplement 2A*) or with the weak base chloroquine (Chlor, 150 µM; *Figure 1E*, *Figure 1—figure supplement 2B*). In *Figure 1E* is a comparison of how varenicline (Var) or lobeline (Lob) affects $^{125}$I-epibatidine binding to α4β2R-expressing HEK cells for all of these conditions. When performed on membrane preparations or in conditions that increased the pH of cellular acidic compartments, $^{125}$I-epibatidine binding was increased consistent with upregulation of α4β2Rs. This upregulation of binding was not observed for intact cells.

The second assay we used for α4β2R upregulation was to measure changes in the mean amplitude of ACh-induced currents in the receptor-expressing HEK cells following exposure to nicotine or other agents (*Figure 2*). Cells were treated for 17–20 hr as shown; we then carried out whole-cell voltage recordings while rapidly applying ACh (1 mM) for 1 s. Nicotine upregulation of currents from α4β2R-expressing HEK cells was observed as an increase by ~5 fold in mean peak current amplitudes (e.g., *Figure 2A*, white vs. black column). Varenicline and lobeline (both at 30 µM) exposure altered the ACh-induced currents in parallel with what we observed in $^{125}$I-epibatidine binding assays with live cells (*Figure 2A,B*). Exposure to varenicline alone did not alter nAChR current amplitudes, whereas lobeline reduced the mean amplitude to below that of control (untreated) receptor currents. When co-incubated with nicotine, both varenicline and lobeline occluded functional upregulation; lobeline treatment again reduced nAChR currents to a mean amplitude below even that of the control group. Neither treatment changed the time course of desensitization of the ACh currents. Thus, varenicline and lobeline prevented nicotine-induced functional upregulation of α4β2R nAChR currents. $NH_4Cl$ (20 mM) treatment also partly restored nicotine-induced functional upregulation of ACh currents in lobeline and varenicline treated α4β2R-expressing HEK cells (*Figure 2C,D*), albeit to a lesser degree than as observed with the binding experiments.

We performed additional experiments to test whether varenicline and lobeline are released from intracellular acidic compartments after being trapped in the α4β2R-expressing HEK cells. The protocol is displayed schematically in *Figure 1—figure supplement 3A*. Briefly, HEK cells chronically treated with varenicline or lobeline for 17 hr were washed to remove free ligands and incubated with $NH_4Cl$-containing PBS for 10 min, which we predicted would cause varenicline or lobeline to be rapidly released from intracellular acidic compartments. The supernatant from the treated cells was then added to a different set of receptor-expressing HEK cells that had not been exposed to varenicline or lobeline. $^{125}$I-epibatidine binding was performed on the second set of cells to test whether each supernatant contained competitive ligands that reduced binding (indicative of the presence of

varenicline or lobeline). Consistent with the release of varenicline and lobeline from intracellular acidic compartments, $^{125}$I-epibatidine binding was highly reduced for cells that received the bathing solution from cells chronically treated with varenicline or lobeline as compared to untreated cells (*Figure 1—figure supplement 3B*). These results strongly suggest that the weak bases re-equilibrate across the plasma membrane from intracellular compartments in NH$_4$Cl, and that they concentrate in sufficiently high concentration to compete for binding to nAChRs.

## nAChR weak base ligands exhibit different degrees of intracellular trapping

We next tested if nicotine itself was trapped in intracellular compartments of HEK cells expressing α4β2Rs. Nicotine upregulation of $^{125}$I-epibatidine binding from live cells was similar to that in membrane preparations (*Figure 1*) and following neutralization of the pH gradient of acidic compartments (*Figure 1D*). Functional upregulation of ACh currents by nicotine also did not change if the pH gradient of acidic compartments was diminished (*Figure 2C,D*). These observations are consistent with previous findings that nicotine is not trapped in the acidic compartment of α4β2R-expressing HEK cells (*Jia et al., 2003*).

We tested whether other α4β2R ligand weak bases were trapped in the intracellular acidic compartment like varenicline and lobeline. Dihydro-beta-erythroid (DHβE, *Figure 3A*) is a weak base and a α4β2R competitive antagonist. Despite being a competitive antagonist, DHβE was found to induce upregulation of α4β2R high-affinity binding sites (*Whiteaker et al., 1998*). Consistent with previous findings, we observed a 2-fold upregulation of $^{125}$I-epibatidine binding with 17 hr of exposure of the cells to 100 µM DHβE (*Figure 3B*). At this concentration, acute application of DHβE completely inhibits ACh-induced currents (data not shown), as was shown previously (*Wu et al., 2006*). Conversely, prolonged DHβE exposure upregulated ACh-induced currents, similar to the binding sites (*Figure 3E*). DHβE did not occlude nicotine upregulation when the two ligands were co-incubated, however, demonstrating that it has a different effect on functional upregulation than varenicline and lobeline. Like nicotine, we see no evidence of DHβE being trapped in binding assays; upregulation of $^{125}$I-epibatidine binding was unchanged whether membranes were assayed (data not shown) or if the pH gradient of acidic compartments was diminished with NH$_4$Cl treatment (*Figure 3B*, green columns in the right half of the graph).

Why do the weak bases nicotine and DHβE differ from varenicline and lobeline with respect to trapping? One likely factor is their degree of protonation at the pH found in intracellular compartments (pH 5–6). Protonation of weak bases depends on the acid dissociation constant, pKa (*Trapp et al., 2008*), and the pKa values of nicotine and DHβE are significantly lower than those of varenicline and lobeline (*Table 1*). Based on this correlation between their physical properties and our current data, we propose that nicotine and DHβE are less likely than varenicline and lobeline to be trapped because they are less likely to be in their protonated form in an acidic compartment.

As another test, we examined how the noncompetitive antagonist mecamylamine (*Figure 3A*), a weak base with a pKa higher than that of varenicline and lobeline (*Table 1*), altered binding or functional activity alone and when co-incubated with nicotine. For exposures for 17–18 hr at concentrations up to 100 µM, there was no evidence of upregulation as assayed by $^{125}$I-epibatidine binding (*Figure 3D*) or by ACh-induced currents (data not shown). Acute application of mecamylamine (100 µM) inhibited ACh-induced currents (data not shown) as previously observed (*Wu et al., 2006*), confirming that the drug had the expected pharmacological activity. Despite its high pKa value, we observed no evidence of mecamylamine being trapped because there was no difference in the binding when performed on membranes (data not shown) or if the pH gradient of acidic compartments was diminished with NH$_4$Cl treatment (*Figure 3D*). We propose that two possible mechanisms could account for the lack mecamylamine trapping and the absence of an effect on nicotine upregulation of nAChR current amplitudes. The ligand has a very high pKa (11.2) and therefore might be in a protonated state and unable to permeate the plasma membrane. However, mecamylamine appears to readily cross the blood-brain barrier and cell membranes (*Shytle et al., 2002*). A more likely possibility is that trapping also requires high-affinity ligand binding to α4β2R located within acidic compartments. The affinity of mecamylamine for its α4β2R binding site is very low (*Table 1*) and thus accumulation is precluded as discussed below.

We predicted that epibatidine itself would be trapped in acidic compartments if indeed a high pKa value (greater than that of nicotine) and high-affinity binding to α4β2R sites are important

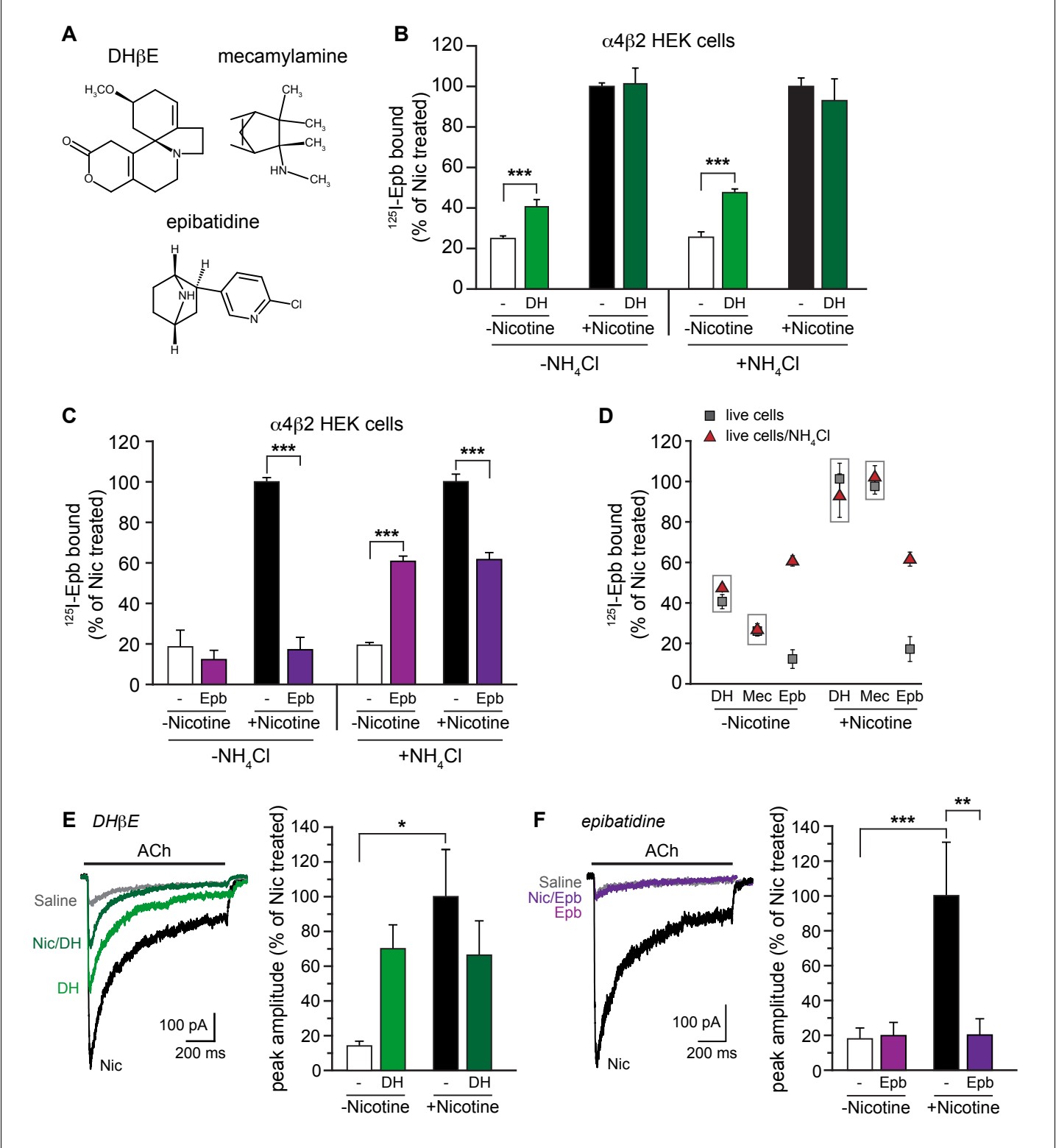

**Figure 3.** α4β2R weak base ligands exhibit different degrees of intracellular trapping. (**A**) DHβE, mecamylamine and epibatidine chemical structures. (**B**) NH₄Cl treatment does not alter DHβE (DH) upregulation. $^{125}$I-epibatidine binding performed on live α4β2R-expressing cells without (left) or with (right) NH₄Cl treatment as in *Figure 1D*. Cells were treated for 17 hr with 100 μM DHβE with or without 10 μM nicotine (n = 6: -NH₄Cl; n = 4: +NH₄Cl)). The profile was similar to cells washed with PBS, indicating no effect of pH on DHβE. (**C**) Intact cells were treated with 30 μM epibatidine (Epb) and $^{125}$I-epibatidine binding were performed after incubating the cells with (left) or without (right) 20 mM NH₄Cl/PBS (n = 3). Column graphs in (**B, C**): ***,

*Figure 3 continued on next page*

*Figure 3 continued*

p<0.001 by one-way ANOVA with Bonferroni's multiple comparison test. n indicates number of independent determinations on separate days and cultures. (D) A distribution plot comparing upregulation ($^{125}$I-epibatidine binding) by DH$\beta$E, mecamylamine and epibatidine before and after disruption of intracellular pH gradient by various agents that raise pH in intact cells. Each point represents the mean and s.e.m from the indicated columns in *Figure 3B,C* with the exception of the mecamylamine data, where the data are not displayed elsewhere, and the points are the means and s.e.m (n = 4) where n indicates number of independent determinations on separate days and cultures. (E) ACh-evoked currents following 17–20 hr treatment of $\alpha$4$\beta$2-expressing HEK cells with vehicle, nicotine, DH$\beta$E (DH, 100 $\mu$M), or nicotine and DH$\beta$E. The number of recordings were 17, 21, 15 and 21, respectively. DH$\beta$E appeared to upregulate ACh function but the currents were variable in their amplitude and the mean was not statistically different from the vehicle group. No attenuation of nicotine upregulation was observed with DH$\beta$E. (F) Epibatidine (Epb, 30 $\mu$M) shows effects on ACh currents similar to that of varenicline. Current amplitudes were similar to the vehicle control when cells were incubated with epibatidine alone, but co-incubation with nicotine prevented functional upregulation. The number of recordings were 14, 22, 17 and 24, respectively. In all the column graphs: *p<0.05; **p<0.01; ***p<0.001 by one-way ANOVA with Tukey's multiple comparison test; n indicates the number of experimental repetitions. Columns show group mean and error bars are the s.e.m.

The following figure supplement is available for figure 3:

**Figure supplement 1.** Dose dependence of epibatidine effects on upregulation.

---

determinants of this phenomenon. The pKa value and affinity for epibatidine are both higher than those for varenicline and lobeline (*Table 1*). In intact cells, epibatidine (30 $\mu$M) did not promote upregulation of either $^{125}$I-epibatidine binding (*Figure 3C* left half, D; *Figure 3—figure supplement 1A*) or ACh-induced currents (*Figure 3F*). However, epibatidine reduced nicotine-induced upregulation in both assays (*Figure 3C* left half, D; *Figure 3—figure supplement 1B*). These effects of epibatidine on binding sites changed when the cells were treated with NH$_4$Cl (*Figure 3C* right half, D; *Figure 3—figure supplement 1A,B*). With NH$_4$Cl treatment, we observed upregulation of $^{125}$I-epibatidine binding to 60% of the nicotine induced value and that nicotine upregulation was reduced by 40%. Thus, epibatidine behaves in a qualitatively similar way as varenicline and lobeline with respect to upregulation and therefore is subject to trapping in intracellular compartments, consistent with the phenomenon arising from a combination of high binding affinity for the receptor and a very basic pKa.

## Direct measurements of $^{125}$I-epibatidine trapping and rate of release

In the preceding experiments, we inferred that trapping in intracellular compartments produced the agonist-specific effects observed in binding and functional assays. We next attempted to directly measure $^{125}$I-epibatidine trapping and release from acidic compartments. We anticipated that intracellular trapping of the radioligand results in two distinct pools of $^{125}$I-epibatidine in the cells: one pool represents $^{125}$I-epibatidine bound to intracellular and cell-surface $\alpha$4$\beta$2Rs, whereas the second

---

**Table 1.** pK$_a$s and K$_i$s for the studied weak base $\alpha$4$\beta$2R ligands

| Weak base | pK$_a$ (Basic) | K$_i$ | Trapping |
|---|---|---|---|
| Varenicline (partial agonist) | 9.2 (*Unal et al., 2012*) | 0.4 nM (*Rollema et al., 2010*) | Yes |
| Lobeline (partial agonist) | 8.8 (*Drugbank, 2016*) | four nM (*Damaj et al., 1997*) | Yes |
| Epibatidine (agonist) | 9.5 (*ChEMBL, 2017*) | 0.01–0.05 nM (*Whiteaker et al., 1998; Badio et al., 1994*) | Yes |
| Nicotine (agonist) | 8.0 (*Barlow and Hamilton, 1962*) | eight nM (*Whiteaker et al., 1998*) | No |
| DH$\beta$E (competitive antagonist) | 7.3 (*ChEMBL, 2017*) | 0.3 $\mu$M (*Whiteaker et al., 1998*) | No |
| Mecamylamine (noncompetitive antagonist) | 11.2 (*Remington and Beringer, 2006; Nangia et al., 1996*) | >1 mM (*Whiteaker et al., 1998*) | No |

pool would arise from unbound $^{125}$I-epibatidine trapped within intracellular acidic compartments. These two pools were differentiated by analysis of the association and dissociation kinetics of $^{125}$I-epibatidine binding in living receptor-expressing HEK cells. First, association rates were determined to test if $^{125}$I-epibatidine binding to intracellular $\alpha4\beta2$Rs is significantly slower than to cell-surface $\alpha4\beta2$Rs, which would be revealed as an association rate curve with two or more rates of binding. However, the data are consistent with a single exponential process and an association rate constant of $3.8 \times 10^7$ (M-sec)$^{-1}$ (*Figure 4—figure supplement 1*), similar to previous measurements using $^3$H-epibatidine on isolated membrane preparations containing $\alpha4\beta2$Rs (*Whiteaker et al., 1998*; *Gnädisch et al., 1999*; *Shafaee et al., 1999*) and to intact cells (*Whiteaker et al., 1998*). The single association rate is consistent with $^{125}$I-epibatidine binding with very similar rates to all cellular $\alpha4\beta2$Rs and indicates that the radioligand crosses cellular membranes rapidly enough that permeation does not significantly contribute to the binding rate (*Figure 4E*).

Second, analysis of the dissociation of $^{125}$I-epibatidine exiting the $\alpha4\beta2$R-expressing cells revealed a biphasic time course that was best fit as the sum of two exponential processes (*Figure 4A*). The faster dissociating component ($\tau$ = 11.4 min; $1.5 \times 10^{-3}$ sec$^{-1}$),~40% of the total binding, was similar to rates previously measured for unbinding of radio-labeled epibatidine from $\alpha4\beta2$Rs in membrane fragments. In contrast, the slower dissociation component ($\tau$ = 47 hr) had a rate constant of $6.0 \times 10^{-6}$ sec$^{-1}$ and therefore was >100 fold slower than previous measurements of $^{125}$I-epibatidine unbinding from $\alpha4\beta2$Rs. We hypothesize that this slow component of $^{125}$I-epibatidine dissociation arose from radioligand trapped in intracellular acidic compartments (illustrated in *Figure 4E*).

The slowly dissociating component in the preceding assays could in part arise from rebinding of $^{125}$I-epibatidine to receptors within intracellular compartments, because epibatidine is a high-affinity agonist and the measured dissociation rate was slow enough to allow repeated cycles of binding and unbinding. We found that 0.10–0.20 pmol of $^{125}$I-epibatidine dissociated during the faster dissociation component in the assay shown in *Figure 4A*, which corresponds to a concentration of 0.2–0.4 nM free $^{125}$I-epibatidine in our 0.5 ml assay volume of the eppendorf tube used in these assays. We predicted that rebinding of radioligand to the receptor should occur at this concentration; to test this hypothesis, we included a membrane-permeable competitive ligand, nicotine (100 µM), at the start of the dissociation measurements. Dissociation of $^{125}$I-epibatidine in the presence of nicotine occurred with a much faster overall time course, such that the radioligand was released in less than a minute rather than being trapped for hours. The rapid release of radioligand was followed by a slower time course of dissociation ($\tau$ = 24 min at 37°C), which is typical of rates ($7.0 \times 10^{-4}$ sec$^{-1}$) previously measured for the dissociation of $^3$H-epibatidine from $\alpha4\beta2$Rs and dissociation constants for affinity ($K_D$ = 18 pM) (*Whiteaker et al., 1998*; *Gnädisch et al., 1999*; *Shafaee et al., 1999*). Thus, rebinding of epibatidine to nAChRs in intracellular compartments contributes to the slowly dissociating pool of radioligand.

We used a high nicotine concentration, 100 µM, to prevent rebinding of $^{125}$I-epibatidine in the experiments shown in *Figure 4B*, but the Ki value for nicotine for $\alpha4\beta2$R binding sites is much lower (8 nM; *Table 1*); in principle, even 100 nM nicotine added during the dissociation measurement should block essentially all rebinding of $^{125}$I-epibatidine. We tested this prediction in additional dissociation assays and instead found that the addition of 100 nM nicotine only reduced the slow component of dissociation by ~20% (*Figure 4C*). This result suggests that rebinding alone does not adequately account for the extraordinarily slow dissociation of $^{125}$I-epibatidine from living cells expressing $\alpha4\beta2$Rs. Rather, two processes are key to the trapping phenomenon: (i) $^{125}$I-epibatidine rebinding to $\alpha4\beta2$Rs and (ii) protonation of $^{125}$I-epibatidine in those same compartments. Higher concentrations of nicotine (100 µM – 10 mM) added into the dissociation assay act on both components of the trapping phenomenon; that is, increasing nicotine concentrations not only prevented $^{125}$I-epibatidine rebinding but also reduced the pH gradient to neutralize acidic compartments, leading to deprotonation (and release) of $^{125}$I-epibatidine. This was evident in the fact that the proportion of $^{125}$I-epibatidine dissociation contributed by the slower component decreased with increasing nicotine concentrations and saturated at a value of ~60% of the total 'bound' $^{125}$I-epibatidine for both the untreated and nicotine-treated cells (*Figure 4C*), which is the entirety of the slowly dissociating component. Chloroquine (150 µM), a weak base with a molecular weight and pKa similar to that of nicotine, also reduces the pH gradient of acidic compartment at similar concentrations and has a similar effect on $^{125}$I-epibatidine binding (*Figure 1E*). Thus, the addition of nicotine during the

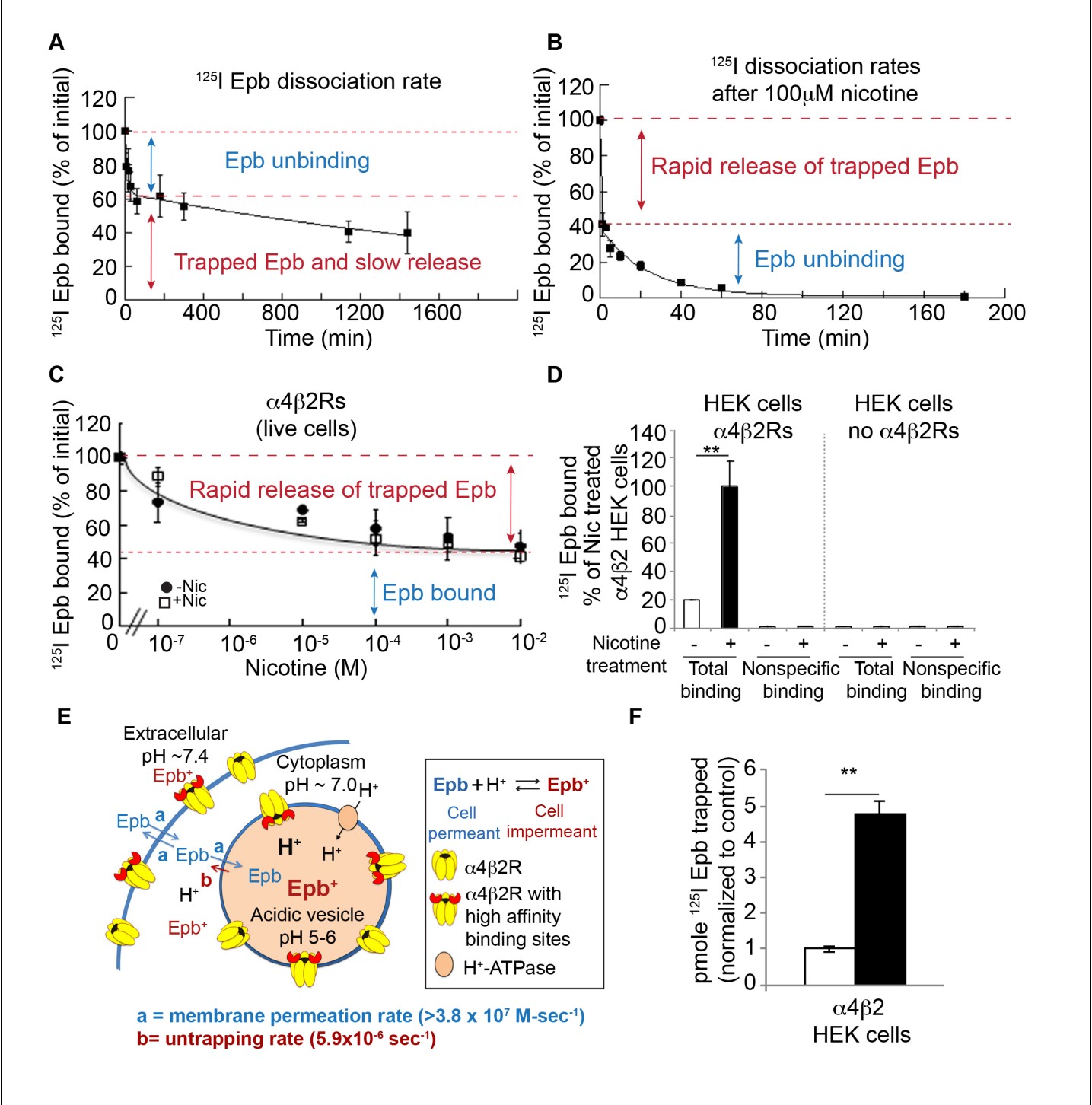

**Figure 4.** Direct measurements of $^{125}$I-epibatidine trapping. (**A**) The biphasic dissociation of $^{125}$I-epibatidine from $\alpha 4\beta 2$R HEK cells measured at 37°C. Bound $^{125}$I-epibatidine is normalized to % bound at time 0. The line through the data represents a least-squares fit of a double exponential function: % Bound(t) = $A_f$ exp($-k_f$t) + $A_s$ exp($-k_s$t) where $A_f$(38 ± 6%) and $A_s$ (62 ± 8%) are the **% bound** $^{125}$I-epibatidine at time (t) = 0 for the fast and slow component respectively; $k_f$ (1.5 ± 0.6×10$^{-3}$ sec$^{-1}$) and $k_s$ (6.0 ± 1.4×10$^{-6}$ sec$^{-1}$) are the time constants for the fast and slow component respectively (n = 4). (**B**) Addition of 100 µM nicotine causes the rapid release of slowly dissociating bound $^{125}$I-epibatidine. Dissociation of $^{125}$I-epibatidine measured at 37°C as in A except after adding 100 µM nicotine at to start the dissociation measurement. The line through the data with the exception of the initial data point represents a least-squares fit of a single exponential function: % Bound(t) = $A_f$ exp($-k_f$t) where $k_f$ was 7.0 ± 1.3×10$^{-4}$ sec$^{-1}$ (n = 3). (**C**) The rapid dissociation of bound $^{125}$I-epibatidine measured after washing the cells with increasing concentrations of nicotine. $\alpha 4\beta 2$R HEK cells were with treated with 10 µM nicotine (dark circles) or left untreated (white squares) for 17 hr. The cells were washed. Cells were washed with PBS followed by

*Figure 4 continued on next page*

Figure 4 continued

three washes with indicated concentrations of nicotine prior to performing $^{125}$I-epibatidine binding (n = 3). For all the points in **A, B, C**: error bar represents mean ± s.e.m. (**D**) $^{125}$I-epibatidine binding to α4β2R-expressing HEK cells versus HEK cells lacking α4β2Rs. 1 mM nicotine was added during $^{125}$I-epibatidine binding to estimate nonspecific binding (n = 3). (**E**) Altered model of ligand trapping with high-affinity α4β2Rs in the acidic vesicles. (**F**) Nicotine exposure increases trapped $^{125}$I-epibatidine. The levels of trapped $^{125}$I-epibatidine were determined with the addition of 100 µM nicotine to start the dissociation. Trapped $^{125}$I-epibatidine was released within 1 min of the nicotine addition as in **Figure 4B and C** for cells untreated (-Nic) or treated with 10 µM nicotine (+Nic) for 17 hr. Plotted is the released $^{125}$I-epibatidine in pmoles normalized to untreated cells (n = 3). For (**D, F**): error bar represents mean ± s.e.m. \*\*p<0.001 by Student's t-test. (**A-D,F**): n indicates number of independent experiments performed on separate days and cultures.

The following figure supplement is available for figure 4:

**Figure supplement 1.** Association rate of $^{125}$I-epibatidine.

dissociation assay serves two functions that together cause rapid release of all $^{125}$I-epibatidine. One function is to bind to α4β2R high-affinity binding sites and block rebinding of $^{125}$I-epibatidine, and the other is to reduce the pH gradient across intracellular compartments in which ligand become trapped.

The preceding experiments strongly suggest that high-affinity α4β2R binding sites are required for trapping of weak base ligands. We tested this idea by assaying $^{125}$I-epibatidine in HEK cells lacking α4β2Rs. $^{125}$I-epibatidine binding to these HEK cells was ~1% of that observed with nicotine upregulation and was equivalent to 'non-specific' binding obtained with 1 mM competing nicotine present during $^{125}$I-epibatidine binding and washing (**Figure 4D**). This finding demonstrates that no selective trapping of $^{125}$I-epibatidine occurs without α4β2R expression. Without high-affinity binding α4β2Rs, $^{125}$I-epibatidine concentrates within acidic vesicles but it rapidly exits the vesicles when its extracellular concentration falls. High-affinity α4β2Rs in intracellular compartments trap $^{125}$I-epibatidine even when its extracellular concentration falls, whereas nicotine rapidly exits the acidic compartments. In this way, the targeting of α4β2Rs to acidic vesicles selectively traps certain weak bases like epibatidine.

Altogether, our data are consistent with the model shown in **Figure 4E**, which depicts how $^{125}$I-epibatidine is selectively trapped within an intracellular acidic compartment. This model differs from that in **Figure 1C** by the addition of high-affinity α4β2Rs in acidic vesicles. In **Figure 4E**, α4β2Rs in the vesicles are depicted in two states, either α4β2Rs with high-affinity binding sites that would contribute to trapping or α4β2Rs without high-affinity binding sites. The two states are consistent with the findings of Vallejo et al. (**Vallejo et al., 2005**) for α4β2Rs at the plasma membrane. When $^{125}$I-epibatidine enters the vesicle lumen, it is protonated and in this state binds to high-affinity sites on α4β2Rs. The high-affinity binding sites appear to play a significant role in selective trapping of $^{125}$I-epibatidine within the vesicle lumen for long periods of time. One pool of radioligand, ~40% of the $^{125}$I-epibatidine, is bound to the surface and intracellular α4β2Rs and dissociates with the characteristic unbinding rate analogous to that measured in membrane preparations. The other pool of epibatidine, ~60%, is not bound to α4β2Rs, but instead is trapped and exits from the cells at a rate much slower than the unbinding rate. As depicted in the model, $^{125}$I-epibatidine exits from acidic vesicles at the slow dissociation rate (b) measured in the experiments shown in **Figure 4A**. As discussed above, the rate that $^{125}$I-epibatidine crosses cellular membranes (a), which is close to diffusion-limited (**Figure 4—figure supplement 1**), is orders of magnitude faster than the dissociation rate (b).

How do the two different pools measured in $^{125}$I-epibatidine dissociation assays change with long-term nicotine exposure? Surprisingly, nicotine upregulation did not change the relative contribution of either the fast or slow component of radioligand dissociation. The fraction of $^{125}$I-epibatidine specifically bound to α4β2Rs remained at ~40% and the fraction of trapped in intracellular compartments and slowly released remained at ~60%. As displayed in **Figure 4F**, nicotine caused the fraction of trapped $^{125}$I-epibatidine to increase by 4–5-fold, the same fold increase observed for total (bound and trapped) $^{125}$I-epibatidine retained in the cells with nicotine exposure (**Figure 4D**). The fraction of $^{125}$I-epibatidine specifically bound to the α4β2Rs also increased 4–5-fold with nicotine exposure (data not shown). This finding is consistent with the results in **Figure 4D** that α4β2Rs

must be present for selective $^{125}$I-epibatidine trapping to occur and further supports the idea that the α4β2Rs are located within intracellular acidic compartments.

## Nicotine exposure redistributes α4β2Rs to acidic vesicles

To address why the trapped $^{125}$I-epibatidine fraction increased by 4–5-fold with nicotine exposure (*Figure 4F*) and to confirm that α4β2Rs are located within intracellular acidic compartments, we made use of a version of an α4 subunit with the pH-sensitive fluorescent tag, super ecliptic pHluorin (SEP) fused to its C-terminus (α4$^{SEP}$; [*Richards et al., 2011*; *Fox et al., 2015*]). SEP is a pH-sensitive variant of GFP that fluoresces at neutral pH but is quenched at pH values lower than 6. When expressed in α4β2R-expressing HEK cells and in cortical neurons, florescent α4$^{SEP}$ was found throughout the ER, as indicated by co-localization with the ER marker, DsRed-ER (*Figure 5—figure supplement 1A,B,C*), and on the cell surface (*Figure 5—figure supplement 1B,D*) as previously observed (*Richards et al., 2011*). To specifically test whether α4$^{SEP}$ subunits were found in intracellular acidic compartments, cells and neurons expressing α4$^{SEP}$ were treated with 20 mM NH$_4$Cl (5 min) to neutralize the pH. The distribution of α4$^{SEP}$ subunit fluorescence was compared before and after NH$_4$Cl treatment.

As displayed in *Figure 5A* for the HEK cells and *Figure 5B* for the cortical neurons, NH$_4$Cl treatment revealed an additional pool of the α4$^{SEP}$ subunits in what appeared to be small vesicles. It is likely that the α4$^{SEP}$ subunits in the acidic vesicles are mature, fully assembled α4β2Rs because receptor assembly occurs in the ER (*Sallette et al., 2005*) and entry into any acidic compartment occurs after exit from the ER. The number of α4β2R-containing acidic vesicles in the HEK cells and cortical neurons was greatly increased by the 17 hr nicotine exposure (*Figure 5A,B,C*) and paralleled the effect of nicotine on the number of high-affinity binding sites and the amount of $^{125}$I-epibatidine trapping (*Figure 4F*). In the neurons, α4β2R-containing acidic vesicles were found in the processes, in what appeared to be both dendrites and axons, as well as in the somata (*Figure 5B*). These results demonstrate that nicotine upregulation occurs in part through an augmentation of the number of receptor-containing acidic vesicles.

## Discussion

In this study, we find that the smoking cessation agent varenicline (Chantix) has significant effects on α4β2Rs independent of its pharmacological action as a partial agonist. Varenicline exposure reduced nicotine upregulation of α4β2Rs, which in vivo causes long-lasting changes in α4β2Rs linked to different components of nicotine addiction (*Govind et al., 2009*; *Vezina et al., 2007*; *Lewis and Picciotto, 2013*). The effects of varenicline on upregulation required intact intracellular acidic vesicles containing α4β2Rs and were not observed if HEK cell or neuronal membranes were compromised. The integrity of acidic compartments was lost in earlier studies that assayed binding using membrane fragments of brain tissue or autoradiography (*Marks et al., 1983*; *Schwartz and Kellar, 1983*; *Marks et al., 2015*) accounting for the different actions of varenicline we observed in living cells. Based on our findings, it is evident that varenicline is selectively trapped as a weak base within intracellular acidic vesicles, and once trapped, it is slowly released from the vesicles and cells. The trapped varenicline competes with $^{125}$I-epibatidine binding to α4β2Rs within the acidic vesicles while the slow release competes with $^{125}$I-epibatidine binding elsewhere in the cells and desensitizes cell-surface α4β2Rs. These effects cause the apparent suppression of binding site and functional upregulation (*Figures 1* and *2*).

We observed effects of varenicline on nicotine upregulation at concentrations as low as 100 nM (*Figure 1G*), a concentration that is estimated to occur in human brain with prescribed varencline doses (*Rollema et al., 2010*). Based on our experimental evidence at higher varencline concentrations, we would expect that the effects of varenicline at 100 nM are caused by the trapping of varenicline in acidic vesicles at this lower concentration. It should also be noted that in order to estimate human brain varenicline concentrations, Rollema *et al* (*Rollema et al., 2010*) used rat brain lysates to measure the high-affinity binding. As is evident from our study, this approach will not include varenicline trapping in acidic vesicles, which should serve as an additional high-capacity reservoir for varenicline in brain neurons, and is lost in the brain lysate preparation. Therefore, this study likely significantly underestimated the levels of varenicline in brain and the effects of trapping in acidic vesicles should be considered in future such estimates.

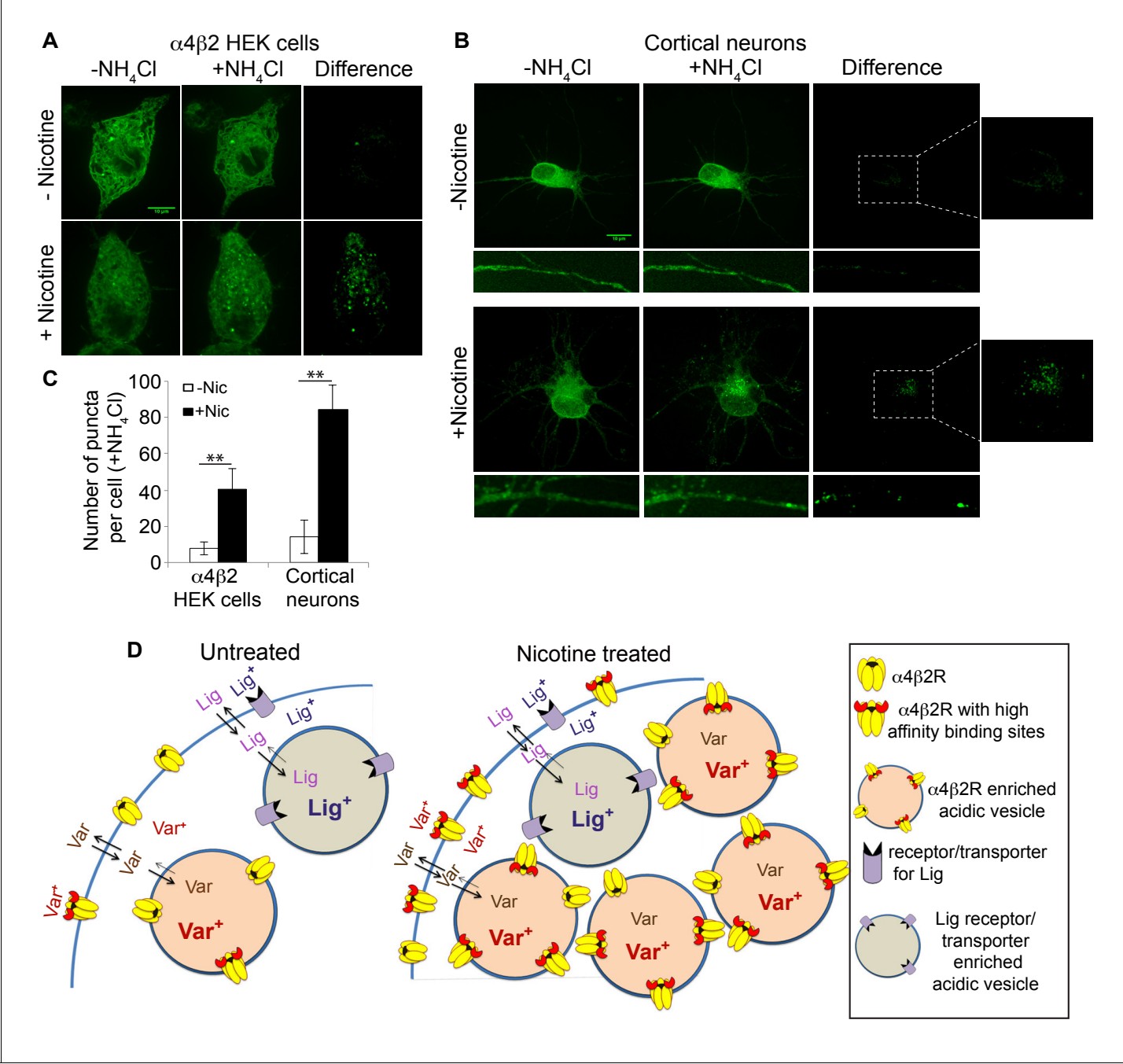

**Figure 5.** Nicotine exposure increased the number of α4β2R-containing acidic vesicles. (**A**) Imaging α4β2R-containing acidic vesicles and the effect of nicotine exposure. α4[SEP] was transfected into the α4β2R-expressing HEK cells. Image (100 x) of 3 merged slices near the cell surface from untreated (top) and nicotine-treated (bottom) cells. Cell was imaged without $NH_4Cl$ (left panel) and after adding $NH_4Cl$ (5 min; middle panel). In the right panel (difference) the total fluorescent intensity in -$NH_4Cl$ image was subtracted from +$NH_4Cl$ image. Scale bar is 10 μm. (**B**) Imaging α4β2R-containing acidic vesicles in cultured neurons and the effect of nicotine exposure. α4[SEP] and β2[HA] subunits was transfected into cortical neurons (DIV 9). Image as in A of untreated (top) and nicotine-treated (bottom) neurons. Both soma and dendrites from the same cells are shown. Right panel and inset shows the difference obtained by subtracting -$NH_4Cl$ image from +$NH_4Cl$ image. (Scale bar:10 μm). (**C**) Quantification of the numbers of acidic vesicles from the difference images. The fluorescent vesicles were counted and plotted for HEK cells (22 cells, for both untreated and nic treated, n = 4) and cortical neurons (10 cells for untreated and 8 cells for nic treated, n = 3). Error bar represents mean ± s.e.m. **p<0.001 by student t test. n indicates number of independent experiments performed on separate days and cultures. (**D**) Model illustrating how trapping in acidic vesicles of weak base ligands like varenicline, lobeline and epibatidine is selective and regulated by nicotine. Ligand trapping is a function of weak base pKa and its affinity of the ligands for α4β2Rs in the case of α4β2R weak base ligands. As illustrated in the figure weak base ligands (Lig) for other receptors or transporters could also be trapped selectively.

*Figure 5 continued on next page*

*Figure 5 continued*

The following figure supplement is available for figure 5:

**Figure supplement 1.** Subcellular distribution of pHluorin tagged α4 subunit (α4$^{SEP}$) in α4β2R-expressing HEK cells and cortical neurons.

Lobeline and epibatidine also appeared to be trapped in intracellular compartments and showed effects similar to that of varenicline (*Figure 1—figure supplement 1C,D* and *Figure 3—figure supplement 1*), whereas nicotine and other weak base ligands with lower pK$_a$s were not significantly trapped and were rapidly washed from cells and cultured neurons. The dose-dependence of all three weak base ligands that were trapped within intracellular acidic vesicles had an inverted U-shape where the peak of the curve occurs at ~1 µM concentration. The decline in the dose-dependence curve at concentrations greater than 1 µM is consistent with the weak base effect of the ligands at these concentrations increasing the pH within the vesicle lumen and thereby increasing the rate of release of the trapped ligand.

Our observation that varenicline, but not nicotine, is trapped in α4β2R-containing acidic vesicles can potentially explain differences in the human pharmacokinetic profiles of these drugs. The decay time (t$_{1/2}$) for the varenicline plasma concentration is 1 day and the time to reach steady-state levels with repeated dosing is 4 days (*Faessel et al., 2010*). In contrast, the decay time for the nicotine plasma concentration to decay is 2 hr and the time to reach steady-state levels with repeated dosing is 2–3 hr (*Benowitz et al., 2009*). These differences are consistent with the slow exit rate we measured for varenicline from acidic vesicles and the rapid exit of nicotine. The residence of a large amount of varenicline in α4β2R-containing acidic vesicles in neurons is likely also to contribute to the differences in the rate at which varenicline and nicotine are metabolized; nicotine is metabolized with the decay time of 2 hr, whereas less than 10% of varenicline is metabolized over this time. The volume of distribution of varenicline and nicotine differs as well. For varenicline, the volume of distribution (5.9 L/Kg; [*Faessel et al., 2010*]) is more than twice that for nicotine (2.6 L/Kg; [*Benowitz et al., 2009*]). This difference again is consistent with and perhaps caused by trapping of varenicline within acidic compartments in cells and neurons and the absence of trapping for nicotine. How varenicline trapping in intracellular acidic vesicles could be altering its clinical efficacy is shown in our working model of this process (*Figures 4E* and *5D*). Based on our findings, the presence of high-affinity α4β2Rs in the vesicles and the weak base nature of varenicline will influence its partitioning between extracellular, cytoplasmic, and vesicular pools. Varenicline trapping in acidic vesicles would create a high-capacity reservoir in neurons that express α4β2Rs. We predict that this phenomenon maintains relatively high and constant concentrations of varenicline, especially in contrast to nicotine levels that rise during the day and decline rapidly at night. Clinical efficacy could therefore result from sustained varenicline levels within neurons that leak out and desensitize α4β2Rs, and perhaps other nAChRs, and that activity counteracts the functional upregulation of nicotine exposure.

As modeled in *Figures 4E* and *5D*, the presence of α4β2R is what provides the selectivity for long-lasting trapping of certain nicotinic weak base ligands in acidic vesicles. Selective trapping also requires a low enough pH in the vesicles, to protonate weak base ligands and slows its exit. Trapping only occurs when there are α4β2Rs in the vesicles and ligand pKa and affinity for α4β2Rs are sufficiently high. Nicotine and DHβE fail to show the trapping phenomenon for two reasons: nicotine and DHβE are not sufficiently protonated at low pH and DHβE also does not bind with high enough affinity to α4β2R. Epibatidine, on the other hand, has a high pK$_a$ and affinity, and therefore accumulates and leaks back out over the course of days, as was observed in our dissociation experiments (*Figure 4A*).

As measured by $^{125}$I-epibatidine (*Figure 4F*), ligand trapping was increased by nicotine exposure to the same degree as the binding to the receptors (*Figure 4D*). Higher trapping levels were caused by a rise in the numbers of vesicles containing α4β2Rs as imaged using pH-sensitive, pHluorin-tagged α4 subunits in the α4β2R-expressing HEK cells and cultured neurons (*Figure 5*). As modeled in *Figure 5D*, the higher numbers of acidic vesicles and the resulting rise in trapping capacity acts as a mechanism to regulate selective trapping of varenicline and other weak base α4β2R ligands. However, it is not clear if the increases are caused by the formation of new vesicles or redistribution of

$\alpha4\beta2$Rs to preexisting vesicles. Intracellular acidic vesicles exist during the late stages of Golgi trafficking and different pools of endosomal membranes, recycling or late endosomes, or lysosomes (*Paroutis et al., 2004*). In this study, we did not distinguish among these possibilities, but our previous findings and those of others (*Vallejo et al., 2005*; *Darsow et al., 2005*) indicate that surface $\alpha4\beta2$R degradation through late endosomes and lysosomes is not increased by nicotine upregulation; therefore, the acidic vesicles are unlikely to be either of these organelles. Still to be determined are which intracellular compartment the vesicles originate from and the role of the vesicles in nicotine upregulation of $\alpha4\beta2$Rs.

If the smoking cessation activity of varenicline is indeed caused by its trapping in intracellular acidic vesicles, then our findings should help guide the design of more effective smoking cessation drugs. While we have established that both the pKa of a ligand and its $\alpha4\beta2$R affinity are important for trapping, further experiments are needed to assay what specific pKa and $\alpha4\beta2$R affinity of a weak base ligand is most effective for smoking cessation. The selective and regulated trapping we observe for varenicline also may occur for weak base ligands that bind with high-affinity to other types of receptors, ion channels or transporters. Most drugs of abuse are weak base ligands that bind with varying affinities to a variety of different membrane proteins (*Sulzer, 2011*). In particular, amphetamines are weak bases with pKa values in the range of 8 to 10 that can concentrate in acidic vesicles and affect the catecholamine quantum size in synaptic vesicles and chromaffin granules (*Sulzer et al., 2005*). Certain antipsychotic drugs are weak bases that accumulate in and are released from synaptic vesicles (*Tischbirek et al., 2012*). As shown schematically in *Figure 5D*, the targeting of other membrane protein high-affinity binding sites to acidic vesicles would allow different classes of weak base ligands to be selectively trapped in those vesicles (Lig in *Figure 5D*). Weak bases that bind with high-affinity to the binding sites where other drugs of abuse bind could serve as cessation agents similar to varenicline.

## Materials and methods

### cDNA constructs: cell culture and transfection

Previously characterized mouse nAChR $\alpha4^{SEP}$ with a super ecliptic pHluorin incorporated on the C-terminus of $\alpha4$ was a gift from Dr. Christopher I. Richards (University of Kentucky, Lexington, Kentucky) (*Richards et al., 2011*). pDsRed-ER was from Clontech Laboratories. Human $\alpha4$ and $\beta2$ cDNA were gifts from Prof. Steven M. Sine (Mayo Clinic, Rochester, Minnesota). Rat $\alpha4$ and $\beta2$ used for generating the stable cell line were provided by Dr. Jim Boulter, University of California, Los Angeles, CA. The HA epitope, YPYDVPDYA, and a stop codon were inserted after the last codon of the 3′-translated region of the subunit DNA of the $\beta2$ using the extension overlap method as described in Vallejo et al (*Vallejo et al., 2005*).

### Cell culture and transfection

The human embryonic kidney (HEK 293T) cell line stably expressing the large T antigen (tSA201 cells) was from Dr. J. Kyle (University of Chicago, Chicago, IL). This cell line is not in the list of Database of Cross-Contaminated or Misidentified Cell Lines. Using this parent HEK 293 T cells, a stable cell line expressing rat $\alpha4\beta2$ nAChRs were generated in our lab and it expresses an untagged $\alpha4$ and a C-terminal HA epitope tagged $\beta2$ subunits (*Vallejo et al., 2005*). Both parent HEK cell line and stable $\alpha4\beta2$ HEK cell line were maintained in DMEM (Gibco, Life technologies) with 10% calf serum (Hyclone, GE Healthcare Life Sciences, UT) at 37°C in the presence of 5% CO2. DMEM was supplemented with Hygromycin (Calbiochem, EMD Millipore, MA) at 0.4 mg/ml for maintaining selection of $\alpha4\beta2$ HEK cells. Transfection of human $\alpha4$ and $\beta2$ subunits into HEK parent cell line or $\alpha4^{SEP}$ into $\alpha4\beta2$ HEK cell line were performed using calcium phosphate method (*Eertmoed et al., 1998*). Stable cells were maintained in hygromycin free DMEM prior to transfection. Hoechst staining and immunofluorescent detection were performed periodically to test for mycoplasma contamination. Fresh batch of cells were thawed and were maintained only upto two months.

Primary cultures of rat cortical neurons were prepared as described earlier (*Govind et al., 2012*). Dissociated cortical cells from E18 Sprague Dawley rat pups were plated on plates that were coated with poly-D-lysine (Sigma, MO). For live imaging, neurons were plated in glass bottom dishes (Mat-Tek, MA). DIV eight neurons were transfected with $\alpha4^{SEP}$, $\beta2_{HA}$ and DsRed ER cDNAs using

lipofectamine 2000 (Invitrogen, Thermofisher scientific, MA) reagent. 2–3 days after transfection, neurons were treated with 1 µM nicotine for 17 hr and were live imaged in low fluorescence Hibernate E buffer (Brain bits, IL).

## Drugs and reagents

Poly-D-Lysine (P7886), Nicotine (N3876), Varenicline (PZ0004), Lobeline (141879), Epibatidine (E1145), Mecamylamine (M9020), Ammonium Chloride (A0171), Chloroquine (C6628) and Bafilomycin A1 (B1793) were purchased from Sigma, MO. Dihydro beta-erythroidine (DH$\beta$E) (2349) was purchased from Tocris, MN. Neurobasal medium, B27, HBSS and DMEM were purchased from Life technologies (Thermofisher scientific, MA).

## $^{125}$I Epibatidine binding

$\alpha4\beta2$ stable cell line was treated with indicated concentrations of nAChR ligands for 17 hr in the presence or absence of 10 µM nicotine. For intact counts, cells were washed four times with PBS, scraped off the plates, pelleted at 2800 rpm for 3 min, resuspended in 1 ml PBS and aliquots were incubated with 2.5 nM $^{125}$I-epibatidine ($^{125}$I Epb) (2200 Ci/mmol; Perkin Elmer) for 20 min at room temperature. At the end of incubation cells are harvested on Whatman GF/B filters presoaked in 0.5% polyethyleneimine and washed four times with PBS using 24-channel cell harvester (Brandel, MD). Non-specific binding was estimated by incubating parallel samples in 1 mM nicotine prior to and during incubation with $^{125}$I Epb. Radioactivity of bound $^{125}$I Epb was determined using gamma counter (Wallac, Perkin Elmer, MA). pmole $^{125}$I Epb bound to cells are normalized to nicotine upregulated cells and plotted as % of nicotine treated cells.

Primary cultures of cortical neurons were treated with indicated concentrations of varenicline or lobeline with or without 1 µM nicotine. Intact neurons were washed three times with PBS, gently scraped off the plates in PBS, pelleted down at 2800 rpm for 3 min and resuspended in PBS prior to binding with 1 nM $^{125}$I Epb. After 20 min of incubation with radioactivity at room temperature, cells are transferred to filters using Brandel cell harvester, filters were washed four times with PBS and radio activity associated with the filters measured as described above. Non-specific binding was calculated from cells incubated with 1 mM nicotine during $^{125}$I Epb binding.

## Disruption of intracellular pH gradient

Intact cells were treated with nAChR ligands for 17 hr and washed twice with PBS. Cells were exposed to two 5 min incubations with PBS containing 20 mM ammonium chloride, 150 µM chloroquine or 50 nM bafilomycin A followed by two PBS washes. Cells were gently scraped off the plates, re-suspended in PBS and subjected to $^{125}$I-epibatidine binding as described above. Depending on the number of samples per experiment, the time elapsed between when the drug-containing medium was removed from the cells and the epibatidine-binding initiated ranged from 25 to 45 min.

## Membrane preparation

$\alpha4\beta2$ HEK cells were treated with indicated concentrations of the drugs with or without 10 µM nicotine for 17 hr. Cells were suspended in hypotonic buffer (10 mM Hepes, pH 7.9, 1 mM MgCl$_2$, 1 mM EDTA) plus protease inhibitors and incubated in ice for 10 min. Cells were homogenized using a dounce homogenizer (10 times). Nuclei were removed by centrifugation at 1000 x g for 10 min. The supernatant was subjected to ultra centrifugation at 100,000 x g for 1 hr at 4°C using Beckman TLA 100.3 rotor. Supernatant was removed and the membrane pellet was resuspended in PBS via sonication. Aliquots of resuspended membrane were bound with 2.5 nM $^{125}$I Epb for 20 min at room temperature and transferred to filter paper and washed using brandel cell harvester.

## Immunofluorescence staining

$\alpha4\beta2$ HEK cells and cortical neurons were plated in glass bottom live imaging plates (MatTek, MA) coated with poly-D-lysine one day before plating. $\alpha4\beta2$ stable cells were transfected with $\alpha4^{SEP}$ with or without pDsRed-ER using calcium phosphate method. After 24 hr of transfection cells were treated 10 µM nicotine for 17 hr. For imaging, cells were incubated with anti-HA (Mouse monoclonal HA.11, Biolegend, CA) antibody for 40 min in DMEM. Cells were washed three times with DMEM and labeled with Alexa Fluor anti-mouse 647 (Molecular Probes, Thermo scientific, MA) secondary

antibody for 30 min. Cells were washed three times with Low Fluorescence Hibernate E and imaged in the same buffer.

Cortical neurons (DIV 8) were transfected with $\alpha4^{SEP}$, $\beta2_{HA}$ and DsRed-ER using lipofectamine 2000 (Invitrogen, Thermo scientific, MA) as per manufacturers protocol. Neurons were treated with 1 µM nicotine either 2–3 days after transfection and imaged after 17 hr of nicotine exposure. Neurons were live labeled with anti HA antibody for 40 min, washed three times followed by secondary antibody Alexa Fluor anti-mouse 647 incubation for 30 min. Antibody incubations and washes were done in conditioned neurobasal medium. Neurons were subsequently washed three times with Hibernate E buffer and imaged in the same buffer.

## Measuring $\alpha4^{SEP}$ in acidic vesicles

SEP is pH sensitive and does not fluoresce at pH below 6. Under physiological pH, the fluorescence emitted from $\alpha4^{SEP}$ is mainly from surface receptors (extracellular SEP) and from receptors in the ER (luminal SEP). The fluorescence of SEP is quenched in intracellular vesicles as their pH is below 6. In order to visualize $\alpha4^{SEP}$ in acidic vesicles, intracellular pH was increased using ammonium chloride ($NH_4Cl$). Cells were initially imaged in Hibernate E buffer (pH 7.4). $NH_4Cl$ was added drop wise to a final concentration of 20 mM with minimum disturbance to the cells. 5–10 min after addition of $NH_4Cl$, confocal images were collected from the same cells again. Three to four consecutive slices of the cells were combined for HEK cells and 7–8 consecutive slices were combined for cortical neurons both for pre and post $NH_4Cl$ images. The integrated density of pre-$NH_4Cl$ image of the cell was subtracted from that of the post-$NH_4Cl$ image. The resultant signal was thresholded so that only puncta that were several folds higher than background were selected. Punctas were evaluated using the Analyze Particles function in ImageJ.

## Electrophysiology

HEK cells stably expressing $\alpha4\beta2$ receptors maintained as described above were plated at low density on glass coverslips before treating for 15–18 hr with media containing nicotine or other drugs as per the protocol for binding experiments. Following the incubation, cells were live-labeled with a 1:200 dilution of mouse anti-HA antibody (EMD Millipore #05–904) for 1 hr at 37°C in media containing the same nicotinic agents. Cells were washed three times with media lacking primary antibody before addition of a 1:500 dilution of the secondary antibody (Thermo-Fisher goat anti-mouse Alexa Fluor 488, catalog # A11029) for 1 hr at room temperature in media containing the same nicotinic agents. Cells were washed three times with media before use in physiology experiments. For $NH_4Cl$ treatment experiments, following the HA labeling the cells were washed twice with PBS and exposed to two 10 min incubations with 20 mM $NH_4Cl$ and several final washes with PBS before use in recordings. Cells were only used for recording a maximum of one hour after completion of the HA labeling or $NH_4Cl$ treatment.

Cells were voltage clamped in whole-cell configuration at a holding potential of $-70$ mV using an Axopatch 200B amplifier running pClamp 10 (Molecular Devices; Sunnyvale, CA). Currents were elicited from cells lifted from the coverslip by the fast application of 1 mM ACh using a piezo-ceramic bimorph system with a solution exchange time of ~1 ms. External solution consisted of (in mM): 150 $NaCl$, 2.8 $KCl$, 1.8 $CaCl_2$, 1.0 $MgCl_2$, 10 glucose, and 10 HEPES, adjusted to pH 7.3. Internal pipette solution was (in mM): 110 $CsF$, 30 $CsCl$, 4 $NaCl$, 0.5 $CaCl_2$, 10 Hepes, and 5 EGTA, adjusted to pH 7.3. Peak amplitudes and time courses of desensitization were determined by *post-hoc* analysis using Clampfit 10. Statistical differences were tested for using one-way ANOVAs with Tukey's Multiple Comparison Test in GraphPad (La Jolla, CA).

### Statistical analyses

All statistical analyses were performed using StatPlus software (AnalystSoft Inc, Walnut, CA), unless otherwise noted. Statistical tests used are indicated in each figure legend.

## Acknowledgements

This work was supported by RO1 DA035430 and a Pilot Project from the University of Chicago Cancer Center. The authors would like to thank Drs. Okunola Jeyifous and Jary Delgado for assistance with rat brain dissections for cortical neuronal preparations, Drs. Steve Sine (human $\alpha4$ and $\beta2$) and

Chris Richards (α4 SEP) for their gifts of nAChR constructs. The authors declare that they have no competing interests.

## Additional information

### Funding

| Funder | Grant reference number | Author |
|--------|------------------------|--------|
| National Institutes of Health | RO1DA 035430 | Anitha P Govind<br>Yolanda F Vallejo<br>Jacob R Stolz<br>Jing-Zhi Yan<br>Geoffrey T Swanson<br>William N Green |
| University of Chicago | Cance center-Pilot grant | Anitha P Govind<br>William N Green |

The funders had no role in study design, data collection and interpretation, or the decision to submit the work for publication.

### Author contributions

APG, Conceptualization, Data curation, Formal analysis, Validation, Investigation, Visualization, Methodology, Writing—original draft, Writing—review and editing; YFV, Data curation, Formal analysis, Validation, Investigation; JRS, J-ZY, Data curation, Formal analysis, Validation; GTS, Conceptualization, Resources, Data curation, Software, Formal analysis, Supervision, Funding acquisition, Validation, Investigation, Visualization, Methodology, Writing—original draft, Writing—review and editing; WNG, Conceptualization, Resources, Data curation, Software, Formal analysis, Supervision, Funding acquisition, Validation, Investigation, Visualization, Methodology, Writing—original draft, Project administration, Writing—review and editing

### Author ORCIDs

Anitha P Govind, http://orcid.org/0000-0002-5890-2395
William N Green, http://orcid.org/0000-0003-2167-1391

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
