## [Decision Letter]

Thank you for submitting your article "Smoking cessation through selective and regulated trapping of nicotinic receptor weak base ligands" for consideration by *eLife*. Your article has been reviewed by two peer reviewers, and the evaluation has been overseen by a Reviewing Editor and Richard Aldrich as the Senior Editor. The reviewers have opted to remain anonymous.

The reviewers have discussed the reviews with one another and the Reviewing Editor has drafted this decision to help you prepare a revised submission.

Summary:

This study addresses the question of why the smoking cessation drug varenicline works, despite the fact that its nature as a strong partial agonist does not account for its long-term actions, and it leads to upregulation of nAChRs much as nicotine does. The work provides evidence that the nature of the compound as a weak base leads it to be internalized and sequestered in acidic vesicles containing α4β4 nAChRs, in a manner that prevents the increase in surface expression of new nicotinic receptors, counteracting the reward-related response associated with addiction.

Essential revisions:

The reviewers found much of interest in the manuscript and their comments included, "This manuscript provides valuable insight into the molecular mechanisms of action of varenicline and points to specific chemical characteristics that could be exploited for the development of new smoking cessation aids. The work is elegant, exhaustive, and significant for the field of nicotine addiction."

They had questions about a number of points however, which require addressing. Most of these have to do with tightening the links between the trapping phenomenon observed and the clinical effect, to strengthen the idea that one is the mechanism underlying the other. The essential revisions are summarized here and expanded below in the reviewers' own words to provide you with their full perspective:

1) Given the manuscript's emphasis on linking the study to clinical use of the drug, more clarification of the relation between drug concentrations tested and clinically relevant concentrations is necessary (reviewers' points 1-4 below).

2) Clarification of the model and the required role of trapping (reviewers' points 5-6 below).

3) Some discussion of the possible mechanism of how receptors are moved to internal compartments would be informative (reviewers' point 7 below).

4) Please edit the title (reviewers' point 8 below).

Reviewers’ points:

1) This manuscript presents results that confirm that intracellular retention of nicotinic ligands can result in a significant residual pool. The residual pool can then bind to nicotinic receptors, at least in intracellular locations. This was first noted (I think) by Whiteaker et al. (op cit, their Methods section) for total binding to cultured cells, and subsequently by Jia et al. (op cit) for surface receptors of *Xenopus oocytes*. The trapping of drugs has been a question of general interest for years and so the application to nicotine dependence is a valuable contribution. However, the authors seek to connect the intracellular retention to clinical effects of smoking cessation drugs (c.f. title of MS). To do so, the authors should relate the experimental observations to the clinically relevant concentrations of drug. For varenicline, the brain concentration is likely to be in the range of 30 to 150 nM (Rollema et al., PMID: 20331614). Based on the data in Figure 1—figure supplement 1, this concentration of varenicline is unlikely to affect upregulation and likely does not result in a significant level of trapped drug. This possibility must be explored. The authors must address the clinically effective concentration in comparison to concentrations of varenicline that show trapping (i.e. NH4Cl effect in Figure 1—figure supplement 1).

2) Similarly the question of species differences could be addressed by briefly discussing Papke et al. (PMID: 20100906) (or repeating the experiments to confirm/deny the difference). The Papke paper noted that varenicline showed a high level of persistent block of rodent (mouse) α4β2 receptors expressed in *Xenopus oocytes*, whereas human receptors showed a significantly lower level. The basis for the difference was not explored, but does this species difference indicate a concern for extension of the present results to behavior in humans?

3) However, I do find it interesting that differences in trapping may result in significant changes in availability of drug as a function of time. Accordingly, it would seem to be imperative that the authors relate their measured time-courses to possible time-courses of brain concentrations for, e.g., nicotine versus varenicline.

4) If trapping is the basis for the clinical efficacy of varenicline, what is the mechanism connecting that to the clinical effect? It seems there are two possibilities: receptors are critical (e.g. that the presence of receptors in acidic vesicles is the mechanism or something to do with rates of partitioning between surface and internal pools) or the time-course of the drug concentration is critical (e.g. the lower level of fluctuation of [varenicline] is important, or the prolonged binding of varenicline to surface receptors, or simply the slower metabolism/clearance of varenicline). Please briefly speculate on the possible mechanism and discuss the relative importance (in their minds) of trapping as compared to the partial agonist nature of the drug in terms of clinical efficacy. It would be quite valuable if the authors provided some data on the effect of varenicline (by itself or in the presence of nicotine) on the level of receptor movement to acidic vesicles (c.f. Figure 5).

5) I do not fully understand the model proposed. From the data presented, lobeline is more slowly removed than nicotine, but perhaps comparably to varenicline. However, the differences in pK and Ki between lobeline and nicotine seem relatively small and so the difference in removal does not seem to be qualitatively expected. Indeed, varenicline would seem to be more different to lobeline than lobeline to nicotine. Am I missing something in the model? In addition to clarifying quantitative points, the authors need to clarify the difference between receptors and "high affinity" receptors shown in the figures and discuss the significance of the two forms.

6) I liked the experiments shown in Figure 4, but they raised a question in my mind. Comparing Figure 4 to 4B, this would suggest that the trapped component (60% of A) is not bound to anything (60% of B). This appears to contradict the apparently essential role of intracellular receptors in promoting trapping. Am I missing something?

7) How does nicotine move nAChRs to acidic vesicles? It would be worth it for the authors to comment about potential mechanisms (i.e. the "chaperone" effect of nicotine) that could be further examined in future studies.

8) The title is a bit "sensational" and could be modified to better reflect the nature of the study.

[Editors' note: further revisions were requested prior to acceptance, as described below.]

Thank you for resubmitting your work entitled "Selective and regulated trapping of nicotinic receptor weak base ligands and relevance to smoking cessation" for further consideration at *eLife*. Your revised article has been favorably evaluated by Richard Aldrich (Senior Editor), a Reviewing Editor, and two reviewers.

The manuscript has been improved but there are some remaining issues that need to be addressed before acceptance, as outlined below:

As you can see from the reviews, pasted below for your reference, there was still some concern about (36) whether clinically relevant concentrations of varenicline are actually trapped, (1) how the high/low affinity receptors figure in to the working model, and (26) whether the working model will be apparent to a reader. In the reviewer consultation, it was agreed regarding point 1 that the simplest interpretation of the observation that 100 nM varenicline reduces 125Epb binding is indeed trapping, and no further experiments are needed.

However, we request that you please make the following additions to the manuscript:

1) Add a line explaining the extrapolation of the effect of 100 nM (clinical concentration) varenicline to the mechanism of trapping.

2) Clarify a little further the proposed role of high vs. low affinity receptors.

3) Incorporate the working model offered in the response to reviewers into the manuscript or give some other indication to the reader of how to conceptualize these data.

*Reviewer #1:*

I found the manuscript easier to read this time (perhaps familiarity). Many of my comments were satisfactorily addressed. There were three that I think still need some work.

The first is the question of brain concentrations. My comment actually read "this concentration of varenicline is unlikely to affect upregulation and likely does not result in a significant level of trapped drug." The authors addressed the first part (upregulation) with new data. However, they did not address the second (the data in Figure 1—figure supplement 1 appears identical to the original). I believe that an essential aspect of the manuscript is that trapping must occur at clinically relevant concentrations, so it is of some concern that this was not addressed. The authors should clarify the data indicating that trapping is significant at 100 nM varenicline.

I also did not completely follow the argument that the brain concentration is likely higher than reported, because trapped drug was not assayed. Of course in the present manuscript the aqueous external concentration is reported, rather than the total intracellular concentration including trapped drug, so the concentrations actually appear to be comparable?

The second is the question of the "high affinity receptors." The single sentence added to the legend (referring to a paper by Vallejo et al) did not clarify the matter for me. Are the postulated low affinity receptors of any relevance to the model? Are numbers and affinity of the low affinity receptors sufficient to provide trapping? What is the affinity postulated for these receptors – that of the 3-α class or weaker? I believe that the "high affinity receptors" are the class of receptor many people regard as the receptor, and occur in two varieties (2 α and 3 α), but this point should be made explicit in the manuscript to avoid confusing readers as to whether high affinity refers to the 2alpha form or not.

The third is the lack of a proposed mechanism. In their response a "working model" is described, but this working model does not seem to be included in the manuscript. The authors should consider succinctly stating the model.

*Reviewer #2:*

The revision has brought clarity and additional supporting data. This is a strong manuscript that provides valuable understanding to the mechanisms of nicotine dependence and smoking cessation therapies.

---

## [Author Response]

*Essential revisions:*

*Reviewers’ points:*

*1) This manuscript presents results that confirm that intracellular retention of nicotinic ligands can result in a significant residual pool. The residual pool can then bind to nicotinic receptors, at least in intracellular locations. This was first noted (I think) by Whiteaker et al. (op cit, their Methods section) for total binding to cultured cells, and subsequently by Jia et al. (op cit) for surface receptors of Xenopus oocytes. The trapping of drugs has been a question of general interest for years and so the application to nicotine dependence is a valuable contribution. However, the authors seek to connect the intracellular retention to clinical effects of smoking cessation drugs (c.f. title of MS). To do so, the authors should relate the experimental observations to the clinically relevant concentrations of drug. For varenicline, the brain concentration is likely to be in the range of 30 to 150 nM (Rollema et al., PMID: 20331614). Based on the data in Figure 1—figure supplement 1, this concentration of varenicline is unlikely to affect upregulation and likely does not result in a significant level of trapped drug. This possibility must be explored. The authors must address the clinically effective concentration in comparison to concentrations of varenicline that show trapping (i.e. NH4Cl effect in Figure 1—figure supplement 1).*

We believe the interesting study by Rollema potentially underestimated the levels of varenicline in brain because the concentrating effect of trapping in acidic vesicles was not considered. Rollema et al. estimated that varenicline concentrations in brain are increased (~4-fold) relative to plasma levels, resulting in their estimated range of 32 to 131 nM, which was derived from both modeling methods and data based on the premise that varenicline concentrations in brain are increased by high-affinity binding sites. Importantly, Rollema et al. considered the influence of high-affinity binding sites in brain lysate studies, which omitted any impact of varenicline trapping because acidic vesicles were lost in these preparations. In our study, we demonstrate that varenicline concentrations will be altered by both high-affinity binding sites and trapping of varenicline in acidic vesicles containing nicotinic receptors.

Regardless of any estimates of varenicline concentration in the CNS, the reviewer makes a valid point that data shown in Figure 1—figure supplement 1 raised some questions regarding whether varenicline concentrations in the range of 30 to 150 nM significantly altered nicotine upregulation. To address the reviewer’s concerns with additional data, we repeated the measurements with 0, 100 nM and 30 μM varenicline. These experiments with rat α4β2Rs were performed side-by-side to compare with the human α4β2R samples to address reviewer’s point 2 (see below). In new Figure 1 nM significantly reduced nicotine upregulation by ~30%; consistent with earlier data, 30 μM varenicline completely reversed upregulation of ^125^I-epibatidine binding by nicotine. This result indicates that varenicline brain levels in the 30 to 150 nM range are indeed likely to affect nicotine upregulation, which was not obvious in our original dataset. In contrast, these concentrations of varenicline will have little to no effect as a partial agonist (Rollema et al., PMID: 20331614).

In summary, we suggest that (i) varenicline concentrations in the CNS are likely to be higher than the range estimated by Rollema et al. (2010) because of weak base trapping in acidic vesicles, and (ii) varenicline suppresses nicotine upregulation at the concentrations estimated by Rollema.

*2) Similarly the question of species differences could be addressed by briefly discussing Papke et al. (PMID: 20100906) (or repeating the experiments to confirm/deny the difference). The Papke paper noted that varenicline showed a high level of persistent block of rodent (mouse α4β2 receptors expressed in Xenopus oocytes, whereas human receptors showed a significantly lower level. The basis for the difference was not explored, but does this species difference indicate a concern for extension of the present results to behavior in humans?*

To address reviewers’ point 2, “the question of species differences”, (see Papke et al. PMID: 20100906) human α4β2Rs were transiently transfected in HEK cells to compare the effect of varenicline on nicotine-induced upregulation of human and rat α4β2Rs. We found no significant differences between rat and human α4β2Rs when nicotine upregulation was measured at 0, 100 nM and 30 mM varenicline (new Figure 1).

*3) However, I do find it interesting that differences in trapping may result in significant changes in availability of drug as a function of time. Accordingly, it would seem to be imperative that the authors relate their measured time-courses to possible time-courses of brain concentrations for, e.g., nicotine versus varenicline.*

*4) If trapping is the basis for the clinical efficacy of varenicline, what is the mechanism connecting that to the clinical effect? It seems there are two possibilities: receptors are critical (e.g. that the presence of receptors in acidic vesicles is the mechanism or something to do with rates of partitioning between surface and internal pools) or the time-course of the drug concentration is critical (e.g. the lower level of fluctuation of [varenicline] is important, or the prolonged binding of varenicline to surface receptors, or simply the slower metabolism/clearance of varenicline). Please briefly speculate on the possible mechanism and discuss the relative importance (in their minds) of trapping as compared to the partial agonist nature of the drug in terms of clinical efficacy. It would be quite valuable if the authors provided some data on the effect of varenicline (by itself or in the presence of nicotine) on the level of receptor movement to acidic vesicles (c.f. Figure 5).*

Points 3 and 4: The reviewers have related concerns in point 3 “it would seem to be imperative that the authors relate their measured time-courses to possible time-courses of brain concentrations for, e.g., nicotine versus varenicline” and part of point 4 “If trapping is the basis for the clinical efficacy of varenicline, what is the mechanism connecting that to the clinical effect? It seems there are two possibilities: receptors are critical (e.g. that the presence of receptors in acidic vesicles is the mechanism or something to do with rates of partitioning between surface and internal pools) or the time-course of the drug concentration is critical (e.g. the lower level of fluctuation of [varenicline] is important, or the prolonged binding of varenicline to surface receptors, or simply the slower metabolism/clearance of varenicline)”.

As far as I know, there are no direct measurements of time-courses for nicotine versus varenicline brain concentrations in rats or humans. The estimates in Rollema et al. (2010) are based on combining diverse measurements – in vivo rat brain concentrations, in vitro rat brain lysates and in vivo human plasma values. Furthermore, as discussed above, the use of in vitro rat brain lysates omits the contribution made by varenicline trapping in α4β2R-containing acidic vesicles, which will be critical for estimating time-dependent changes in varenicline brain concentrations. Thus, it is not possible at this time to address point 3, although we hope to gain insight in future human imaging studies that are just being initiated.

Point 4 (cont.): The question of the possible mechanism of clinical efficacy is one that we agree is very interesting, and we proposed that acidic vesicle trapping and subsequent suppression of upregulation is a reasonable alternate to a hypothesis predicated on the partial agonist activity of varenicline. Our data shows that, unlike nicotine, varenicline becomes selectively trapped in α4β2R-containing acidic vesicles because of its high pKa value and high-affinity for α4β2Rs. Our working model for how this process relates to clinical efficacy incorporates several parameters identified by the reviewer. Certainly, α4β2Rs are required, as our data in Figure 4 shows, and the weak base nature of varenicline will influence partitioning between extracellular, cytoplasmic, and vesicular pools. Varenicline trapping in acidic vesicles creates a high-capacity reservoir in neurons that express α4β2Rs. We predict that this phenomenon maintains relatively high and constant concentrations of varenicline, especially in contrast to nicotine levels that rise during the day and decline rapidly at night. Clinical efficacy could therefore result from sustained varenicline levels within neurons that leaks out and desensitizes α4β2Rs, and perhaps other nAChRs, and that activity counteracts the functional upregulation of nicotine exposure. Our data demonstrate that each of these processes can occur; what remains speculative is whether they do indeed underlie the smoking cessation efficacy of varenicline in humans. We hope that our data will stimulate studies designed to discriminate between this new hypothesis and that based on the pharmacological activity of varenicline.

Point 4 (cont.): “It would be quite valuable if the authors provided some data on the effect of varenicline (by itself or in the presence of nicotine) on the level of receptor movement to acidic vesicles (c.f. Figure 5).”

We agree that this would be of value. However, these experiments are very time consuming and would have prevented resubmission within a 2-month time frame. We do not believe that they would add much more conceptually than what is already presented in Figure 5 with upregulation by nicotine. Experiments are now in progress as part of a large follow-up study to understand how α4β2Rs are trafficked into acidic vesicles and how this is affected by nicotine, varenicline, lobeline and other compounds.

*5) I do not fully understand the model proposed. From the data presented, lobeline is more slowly removed than nicotine, but perhaps comparably to varenicline. However, the differences in pK and Ki between lobeline and nicotine seem relatively small and so the difference in removal does not seem to be qualitatively expected. Indeed, varenicline would seem to be more different to lobeline than lobeline to nicotine. Am I missing something in the model?*

We propose that how the diverse ligands impact nicotine upregulation is dependent upon both pKa and affinity for α4β2Rs. The pKa of lobeline is almost a full unit (or an order of magnitude on a non-log scale) greater than that for nicotine. On the other hand, the Ki value of lobeline is only twice that of nicotine. The difference between lobeline and nicotine actions therefore arises primarily because of the substantial divergence in extent of protonation in acidic vesicles. This and data with other ligands led to our conclusion that both features, pKa and Ki, are critical for determining if trapping occurs.

*In addition to clarifying quantitative points, the authors need to clarify the difference between receptors and "high affinity" receptors shown in the figures and discuss the significance of the two forms.*

We apologize for this omission. The two types of α4β2Rs depicted in the models portrayed in Figure 1, Figure 4 and Figure 5, are based on our findings described in Vallejo et al. (2005) (PMID:15944384), in which we found that biotinylated α4β2Rs on the cell surface form high-affinity binding sites with subsequent nicotine exposure (see also Henderson and Lester PMID: 25660637). This result demonstrated that prior to nicotine exposure there are two populations of α4β2Rs, one population with high-affinity sites and a second populations without high-affinity sites. After nicotine exposure the population with high-affinity sites increases. We have added an explanation about this finding to the figure legend of Figure 1 to explain the difference between the two α4β2R forms.

*6) I liked the experiments shown in Figure 4, but they raised a question in my mind. Comparing Figure 4 to 4B, this would suggest that the trapped component (60% of A) is not bound to anything (60% of B). This appears to contradict the apparently essential role of intracellular receptors in promoting trapping. Am I missing something?*

The reviewer is correct that 60% of what has been interpreted as binding with ^125^I-epibatidine binding is actually not bound to α4β2Rs but instead trapped within the acidic vesicles. However, this *does not* contradict the apparently essential role of α4β2Rs with their binding sites within the lumen of the acidic vesicles receptors in promoting trapping of epibatidine, lobeline and varenicline. As we discuss in the manuscript, selective trapping of these weak base ligands requires high-affinity binding together with a high enough pKa value that the ligands remain protonated within the lumen of the acidic vesicles. The positively charged ligands go through cycles of binding and unbinding, which promote their retention within the vesicles.

*7) How does nicotine move nAChRs to acidic vesicles? It would be worth it for the authors to comment about potential mechanisms (i.e. the "chaperone" effect of nicotine) that could be further examined in future studies.*

We agree with the reviewer that trying to understand how nicotine exposure increases the number of acidic vesicles is a very important issue that needs further investigation, especially with respect to characterizing the underlying mechanisms. We noted in the Discussion, however, that in our opinion the critical issues to be resolved first are whether “the increases [in acidic vesicles] are caused by the formation of new vesicles or redistribution of α4β2Rs to preexisting vesicles” and resolution of the nature of intracellular acidic vesicles that contain α4β2Rs. Addressing these questions will narrow down the number of potential mechanisms. For example, the chaperone effect mentioned by the reviewer is typically associated with movement of α4β2Rs from the ER to the Golgi (Henderson and Lester PMID: 25660637), the site of generation of acidic vesicles. An alternate possibility is that α4β2Rs on the cell surface are endocytosed and enter either a recycling pool of endosomes or late endosomes and lysosomes for degradation. It seems very likely that the mechanisms moving α4β2Rs to acidic vesicles are different for endocytosed α4β2Rs then for α4β2Rs moving from the ER to Golgi.

*8) The title is a bit "sensational" and could be modified to better reflect the nature of the study.*

The title has been changed to “Selective and regulated trapping of nicotinic receptor weak base ligands and relevance to smoking cessation”.

[Editors' note: further revisions were requested prior to acceptance, as described below.]

*The manuscript has been improved but there are some remaining issues that need to be addressed before acceptance, as outlined below:*

*As you can see from the reviews, pasted below for your reference, there was still some concern about (36) whether clinically relevant concentrations of varenicline are actually trapped, (1) how the high/low affinity receptors figure in to the working model, and (26) whether the working model will be apparent to a reader. In the reviewer consultation, it was agreed regarding point 1 that the simplest interpretation of the observation that 100 nM varenicline reduces 125Epb binding is indeed trapping, and no further experiments are needed.*

*However, we request that you please make the following additions to the manuscript:*

*1) Add a line explaining the extrapolation of the effect of 100 nM (clinical concentration) varenicline to the mechanism of trapping.*

To address this concern we have added the following: “We observed effects of varenicline on nicotine upregulation at concentrations as low as 100 nM (Figure 1), a concentration that is estimated to occur in human brain with prescribed varenicline doses (Rollema et al., 2010). Based on our experimental evidence at higher varenicline concentrations, we would expect that the effects of varenicline at 100 nM are caused by the trapping of varenicline in acidic vesicles at this lower concentration.”

2) Clarify a little further the proposed role of high vs. low affinity receptors.

We now state: “In Figure 4, α4β2Rs in the vesicles are depicted in two states, either α4β2Rs with high-affinity binding sites that would contribute to trapping or α4β2Rs without high-affinity binding sites. […] The high-affinity binding sites appear to play a significant role in selective trapping of ^125^I-epibatidine within the vesicle lumen for long periods of time.”

*3) Incorporate the working model offered in the response to reviewers into the manuscript or give some other indication to the reader of how to conceptualize these data.*

To address this concern we have added the following: “How varenicline trapping in intracellular acidic vesicles could be altering its clinical efficacy is shown in our working model of this process (Figure 4 and Figure 5). […] Clinical efficacy could therefore result from sustained varenicline levels within neurons that leaks out and desensitizes α4β2Rs, and perhaps other nAChRs, and that activity counteracts the functional upregulation of nicotine exposure.”